# SIDiffAgent: Self-Improving Diffusion Agent

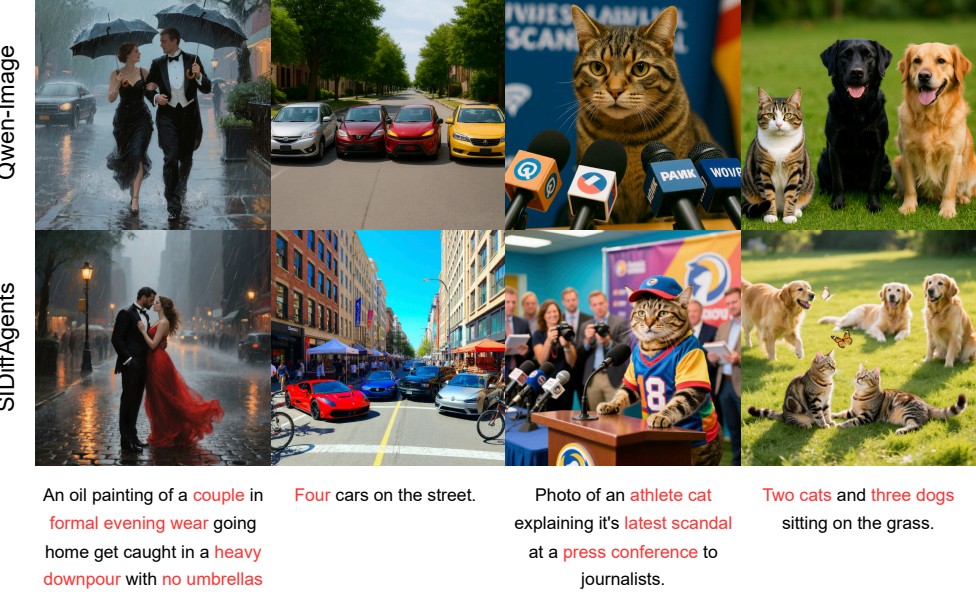

Figure 1: Qualitative comparison between our model, *SIDiffAgent*, and the state-of-the-art, Qwen-Image model. *SIDiffAgent* demonstrates superior text-to-image alignment and realism, accurately rendering prompts that require nuanced understanding. Notably, our model successfully handles: negative constraints ('no umbrellas'), complex scene composition, anthropomorphic roles ('athlete cat'), and precise object counting.

## Abstract

Text-to-image diffusion models have revolutionized generative AI, enabling high-quality and photorealistic image synthesis. However, their practical deployment remains hindered by several limitations: sensitivity to prompt phrasing, ambiguity in semantic interpretation (e.g., "mouse" as animal vs. a computer peripheral), artifacts such as distorted anatomy, and the need for carefully engineered input prompts. Existing methods often require additional training and offer limited controllability, restricting their adaptability in real-world applications. We introduce Self-Improving Diffusion Agent (*SIDiffAgent*), a training-free agentic framework that leverages the Qwen family of models (Qwen-VL, Qwen-Image, Qwen-Edit, Qwen-Embedding) to address these challenges. *SIDiffAgent* autonomously manages prompt engineering, detects and corrects poor generations, and performs fine-grained artifact removal, yielding more reliable and consistent outputs. It further incorporates iterative self-improvement by storing a memory of previous experiences in a database. This database of past experiences is then used to inject prompt-based guidance at each stage of the agentic pipeline. *SIDiffAgent* achieved an average VQA score of 0.884 on GenAIBench, significantly outperforming open-source, proprietary models and agentic methods. We will publicly release our code upon acceptance.

# 1 INTRODUCTION

Image generation models (Rombach et al., 2021; Ramesh et al., 2022; Saharia et al., 2022a; Singer et al., 2022; Ho et al., 2022; Blattmann et al., 2023; Esser et al., 2023) have transformed digital image synthesis, enabling the creation of high-quality visuals with remarkable detail and realism from simple text descriptions. Despite these advances, the performance of state-of-the-art text-to-image diffusion models remains highly sensitive to the precise phrasing of input prompts, making them unable to capture the true intent of the user and creating a significant "intent gap" between the user's intended meaning and the textual instructions provided to the model (Chen et al., 2025a). Here, intent refers to the underlying concept or goal that the user wishes the model to capture. This gap arises because natural language prompts are often ambiguous, underspecified, or casually phrased, leading the model to generate outputs that deviate from the user's actual goal. For instance, the prompt "a mouse" may yield different outputs, either an animal or a computer peripheral, depending on how the model interprets the text.

Another key challenge lies in the distributional mismatch between the unstructured, free-form language typical of end-user prompts and the highly curated, descriptive captions created by human experts and subjected to intensive filtering during the training of diffusion models (Li et al., 2024d; Wu et al., 2025a; Labs, 2024). This discrepancy often results in inadequate textual guidance for generation. For instance, prompts are frequently underspecified (e.g., "a car on a road" without details on the car's model, color, or environment), which forces the model to make unguided assumptions. Such underspecification not only compromises output quality but also increases computational costs, as users must repeatedly regenerate outputs to achieve satisfactory results. These issues constitute a critical bottleneck for the scalability and reliability of diffusion models in professional domains such as digital marketing, content creation, and design, where robust and predictable alignment with user intent is essential.

To mitigate these challenges, prior work such as T2I-Copilot (Chen et al., 2025a) and MCCD (Li et al., 2025a) have proposed agentic frameworks that address ambiguities through image editing and agentic coordination. However, these approaches often lack fine-grained controllability and robust mechanisms for post-generation artifact correction. A key challenge in such Multi-Agent Systems is coordination. Agents often have only partial observability and limited knowledge of others capabilities or the effects of their own actions, leading to suboptimal outcomes (Cemri et al., 2025). Recent works in the field of LLM Agents address this via Theory of Mind (ToM), the ability to attribute goals and beliefs to others (Cross et al., 2024; Yang et al., 2025b). With ToM, agents can understand peers intentions and anticipate future actions, enabling better coordination. To the best of our knowledge, no work has applied this to diffusion-based agents.

Additionally, ReNeg (Li et al., 2025b) highlights the role of negative prompts and embeddings (Ho & Salimans, 2022) in constraining diffusion processes and guiding models away from undesired artifacts. In this work, we introduce a training-free, multi-agent framework *SIDiffAgent* that creates a robust and self-improving Theory of Mind inspired image generation pipeline. *SIDiffAgent* leverages the capabilities of the Qwen model family (Qwen-VL (Bai et al., 2025), Qwen-Image (Wu et al., 2025a), Qwen-Edit (Wu et al., 2025a), Qwen-Embedding (Zhang et al., 2025)) as the backbone of our agentic flow. Our framework uses multiple agents to refine prompts, reuse past experience, and fix artifacts with local edits. Further building upon the observations from ReNeg (Li et al., 2025b), *SIDiffAgent* also incorporates an additional adaptive negative prompt generation agent. Beyond controllability and artifact mitigation, we emphasize the importance of self-improvement: unlike prior agentic frameworks, *SIDiffAgent* is designed to leverage past successes and failures to iteratively refine its generation strategies. While our contribution focuses on system-level architecture rather than a new training objective, we highlight three technically novel components. First, *SIDiffAgent* introduces the first experience-driven memory mechanism for diffusion-based agents, enabling retrieval-conditioned guidance throughout the generation workflow. Second, our Theory-of-Mind–inspired inter-agent reasoning models predictive expectations over other agents' behaviors using accumulated success/pitfall patterns, providing dynamic workflow guidance beyond prompt optimization. Third, SIDiffAgent offers a training-free alternative to RL-based or reward-guided finetuning approaches (e.g., DPO-Diffusion, ReNeg), enabling practical quality improvements even for closed-weight diffusion systems without any retraining.

We empirically validate the effectiveness of *SIDiffAgent* across multiple benchmarks, including GenAIBench and DrawBench, where it establishes new state-of-the-art performance. Our approach not only surpasses prior agentic systems such as T2I-Copilot, achieving a VQA Score (Lin et al., 2024) improvement of 8.73%, but also outperforms leading proprietary models (e.g., Recraft V3, Imagen 3, Flux 1.1-Pro), delivering a 5.36% improvement over Imagen 3. Furthermore, it significantly exceeds the performance of powerful open-source systems (e.g., Flux.1-Dev, HunyuanDiT v1.2, and SD 3.5), achieving a 15.70% improvement over SD 3.5.

Our key contributions are summarized as follows:

- We introduce a multi-agent system that refines the inputs to a diffusion model at test time, ensuring consistent understanding of prompts even when they are expressed in different ways. This results in generations that more faithfully capture the user's intended concepts.

- We adapt a structured image editing approach that enables targeted, incremental corrections to generated images without requiring complete regeneration, narrowing the intent gap and improving alignment between user goals and model generations.

- We build an iterative self-improvement mechanism in which an agentic memory tracks successes, artifacts and pitfalls of various agents, and using them to inject corrective guidance based on Theory of Mind. This creates a training-free, self-improving agentic framework that progressively enhances generation quality over time.

## 2 RELATED WORKS

### 2.1 AI AGENTS

Agents are autonomous entities capable of perceiving their environment, reasoning over goals, and executing actions, often collaboratively through structured memory, planning, and tool use (Luo et al., 2025). In commercial applications such as browser automation or coding (Hong et al., 2024; Qian et al., 2023), these agents typically rely on a large language model (LLM) backbone, which performs reasoning over the current state to determine the optimal action. An extension of this paradigm is the multi-agent framework, in which multiple agents collaborate not only to reason over their own states but also to reason about the states of other agents (Li et al., 2023). Building on these foundations, researchers have started exploring agentic workflows for image generation. In such systems, multi-agent pipelines decompose and refine prompts, evaluate outputs, and correct artifacts. Examples include T2I-Copilot (Chen et al., 2025a), which integrates prompt interpretation, model selection, and quality evaluation, and PromptSculptor (Xiang et al., 2025), which iteratively transforms vague user prompts into precise ones. These systems demonstrate how agentic principles can reduce the intent gap in text-to-image generation, improving fidelity, controllability, and efficiency without requiring additional model training.

### 2.2 THEORY OF MIND IN AGENTIC SYSTEMS

Building on multi-agent systems, Theory of Mind (ToM) (Rabinowitz et al., 2018; Rocha et al., 2023) has become an increasingly important concept for designing smarter and more adaptive agents. It allows agents to understand, infer, and reason about what other agents might believe, desire, or intend in a given context. Recent research in multi-agent and conversational settings has shown that endowing agents with this capability significantly improves both overall system performance and individual agent effectiveness (Yang et al., 2025a; Gu et al., 2025; Jim & Giles, 2001). To the best of our knowledge, this idea has not yet been systematically applied to diffusion-based generative models. In *SIDiffAgent*, ToM is implemented through specialized sub-agents that continuously track each other's mental states, typical successes, potential pitfalls, and characteristic behavior patterns. This information is integrated with a self-improvement mechanism that learns from past experiences with similar inputs to dynamically update these states over time. As a result, each sub-agent can proactively anticipate potential pitfalls and successes of other agents, adjusting its own reasoning accordingly. This awareness allows agents to make decisions that improve not only their individual output quality but also the collective performance of the other sub-agents, ultimately leading to a more robust, adaptive, and coordinated agentic system.

## 2.3 Prompt Optimization for Diffusion Models

Prompt Optimization (PO) has emerged as a crucial technique for aligning large language models (LLMs) with human intent and improving downstream task performance. Originating from in-context learning, PO has evolved into automatic prompt engineering, where LLMs iteratively refine and adapt prompts to achieve more precise and contextually appropriate outputs. While PO has been extensively studied in the context of LLMs, its application to diffusion models remains comparatively underexplored, despite the growing importance of text-to-image generation. For instance, Promptist (Hao et al., 2023) fine-tunes an LLM via reinforcement learning from human feedback to augment user prompts with stylistic and artistic modifiers, while OPT2I (Mañas et al., 2024) leverages an LLM to refine prompts directly for higher-quality image synthesis. Other works, such as DPO-Diffusion (Wang et al., 2024a) and ReNeg (Li et al., 2025b), focus on optimizing negative prompts, either through LLM reasoning or learned embeddings that guide the generative process away from undesired features. TextGrad (Yuksekgonul et al., 2024) introduces gradient-based text feedback optimization, and manual strategies for prompt engineering have also been systematically investigated. Our agentic system builds on these ideas by jointly optimizing positive and image-specific negative prompts at test time via prompt engineering, dynamically balancing guidance to produce more reliable, semantically aligned, and visually coherent images without requiring additional model training.

## 2.4 Self-Improving AI

Recent work has also explored self-improving models and agents, where systems enhance their own performance via interaction or feedback. For LLMs, approaches such as LMSI (Huang et al., 2024) leverage reward signals for autonomous improvement, while IPO (Garg et al., 2025) and Self-Rewarding (Yuan et al., 2024) LLMs optimize through preference pair construction. Methods like SPPO (Wu et al., 2024b) apply self-play to iteratively refine capabilities. Beyond LLMs, agentic systems such as MaestroMotif (Klissarov et al., 2024) use feedback-driven skill acquisition, and AgentSquare (Shang et al., 2024) abstracts agent design into modular components for automated exploration. Other works, such as AlphaEvolve (Novikov et al., 2025) and ADAS (Hu et al., 2025), employ meta-agents to improve task-specific agents, while SICA (Robeyns et al., 2025) eliminates the meta/target distinction by enabling agents to directly edit their own codebases. While the field of self-improving LLMs and coding agents has been actively studied, diffusion-based agents have not been explored in this context. To the best of our knowledge, we present the first training-free, self-improving diffusion framework, where agentic memory and prompt engineering enable self-improvement without retraining.

## 3 Proposed Method: Self-Improving Diffusion Agent (*SIDiffAgent*)

Our framework is built around an agentic workflow that coordinates prompt engineering, image generation, evaluation, and self-improvement. This workflow is orchestrated by three primary agents, the Generation Orchestrator Agent ($A_{\text{ORC}}$), which preprocesses user prompts through four sub-agents, the Creativity Analysis Sub-Agent ($S_{\text{CRE}}$), Intention Analysis Sub-Agent ($S_{\text{INT}}$), Prompt Refinement Sub-Agent ($S_{\text{REF}}$), and Adaptive Negative Prompt Sub-Agent ($S_{\text{NEG}}$) and Generation Sub-Agent ($S_{\text{GEN}}$) for the final generation. The Evaluation Agent ($A_{\text{EVAL}}$) assesses each output, triggering editing when necessary. Finally, the Guidance Agent ($A_{\text{GUID}}$) facilitates learning from past experiences. The overall workflow of *SIDiffAgent* is depicted in Figure 2 and the algorithm is given in Appendix J. Each agent is discussed in detail below.

### 3.1 Generation Orchestrator Agent ($A_{\text{ORC}}$)

The Generation Orchestrator Agent forms the foundation of our approach. It performs a structured preprocessing sequence that balances user intent, creative enrichment, and model compatibility. To achieve this, it operates through five specialized sub-agents, each addressing a distinct aspect of prompt engineering and generation. The following is a detailed explanation of all the agents. The prompts for each of the agents are in the Appendix I.1.

**Step 1. Creativity Analysis Sub-Agent ($S_{\text{CRE}}$) :** The first step is to estimate the freedom that the agents can have which guides the intensity of refinements in the later stages. It classifies the

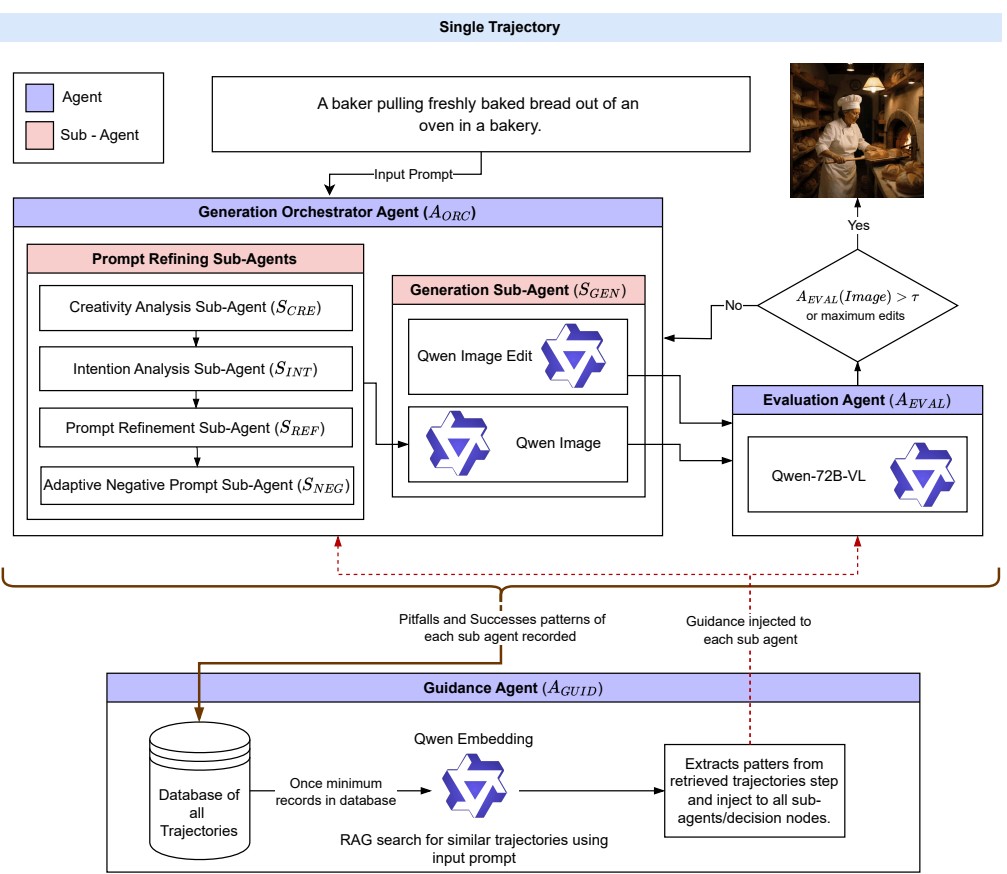

Figure 2: Workflow of the Self-Improving Diffusion Agent (*SIDiffAgent*). An input prompt is processed by the Generation Orchestrator Agent ($A_{\text{ORC}}$), which employs sub-agents ($S_{\text{CRE}}$, $S_{\text{INT}}$, $S_{\text{REF}}$, $S_{\text{NEG}}$) to assess creativity, clarify intent, refine the prompt, and add adaptive negative constraints before generation ($S_{\text{GEN}}$). The Evaluation Agent ($A_{\text{EVAL}}$) scores the generated image on aesthetic quality and text–image alignment, triggering targeted refinements if the evaluation score is less than the pre-defined threshold $\tau$. Each trajectory is stored in the knowledge base, where the Guidance Agent ($A_{\text{GUID}}$) stores pitfalls and successes into corrective and workflow guidance, which are injected back into decision nodes to improve future generations.

input into three levels: high for short or vague prompts, requiring substantial enrichment; medium for moderately detailed prompts, needing balance between structure and interpretation; and low for highly detailed prompts, where strict adherence is key. For instance, the prompt 'a cat' is classified as high creativity, (e.g., specifying breed, pose, or environment), whereas the prompt 'a Persian cat sitting on a red sofa in a living room' would be classified as low creativity, since it already provides detailed constraints. Figure 6 showcases the prompts and image generation pairs for different creativity levels.

**Step 2. Intention Analysis Sub-Agent ($S_{\text{INT}}$):** After establishing the creativity level, this sub-agent deconstructs the users input to produce a comprehensive semantic layout of the prompt. The process extracts core content from the prompt, such as *main subjects*, their *attributes*, their *spatial relationships*, *background*, and additional parameters like *composition*, *lighting*, and overall *visual style*.

Beyond structural parsing, another critical function of this agent is to identify and resolve ambiguities or missing information within the prompt conditioned on the creativity level determined by ($S_{\text{CRE}}$). For prompts designated as low creativity, the agent generates minor assumptions. Conversely, for high creativity prompts, it autonomously assumes information while adhering to the input prompt and ensuring coherent and contextually appropriate details. The output of this stage is

a structured specification that formalizes the user's intent and generates clarifications. For example, given the prompt 'a car' at low creativity, $S_{\text{INT}}$ preserves the minimal description without adding details. At high creativity, however, it expands the prompt by generating clarifying attributes such as the car model, color, background, etc. These details are autonomously inferred by $S_{\text{INT}}$ to maintain coherence while enriching the original input.

**Step 3. Prompt Refinement Sub-Agent ($S_{\textbf{REF}}$):** Building upon the semantic layout generated from $S_{\text{INT}}$, $S_{\text{REF}}$ produces a clearer and refined version of the prompt. $S_{\text{REF}}$ ensures fidelity to the user's intent by preserving all original subjects and scene elements, while simultaneously restructuring the prompt into a more contextually rich and coherent form.

**Step 4. Adaptive Negative Prompt Sub-Agent ($S_{\textbf{NEG}}$):** In classifier-free guidance in diffusion models, the model normally contrasts the conditional prompt with an unconditional prompt (Ho & Salimans, 2022). Negative prompts are injected by replacing that unconditional branch with the negative text embedding, so the guidance steers outputs away from undesired features. This provides fine-grained control, enabling the explicit exclusion of unwanted objects, styles, or quality defects. While the refined prompt captures what should appear in the image, an equally important step is to constrain what must not appear.

$S_{\text{NEG}}$ first appends universal safeguards to mitigate common artifacts (e.g., 'low quality, blurry, distorted, watermark'). Then it analyzes the refined prompt to derive semantically relevant, scene-specific negations. For instance, a prompt specifying a "clear blue sky" adds "clouds, dark clouds" to the negative prompt, while "a person standing alone" generates an exclusion for "extra people or crowd". These adaptive constraints are concise, context-aware, thereby improving output quality and adherence without undermining user intent.

**Step 5. Generation Sub-Agent ($S_{\textbf{GEN}}$):** The final component in $A_{\text{ORC}}$ is the generation sub-agent, which is responsible for the task of generating the image. This sub-agent has 2 models (Qwen-Image and Qwen-Image-Edit) to handle both initial creation and subsequent refinement. Initially, it employs Qwen-Image to generate an image from the refined positive and adaptive negative prompts. Following this step, the evaluation is initiated (Discussed in Section 3.2). If $A_{\text{EVAL}}$ identifies defects or misalignments, $S_{\text{GEN}}$ invokes Qwen-Image-Edit, a specialized editing model, to perform targeted corrections. This strategy of combining generation with precise editing allows the system to iteratively improve the output, ensuring higher fidelity to complex user specifications. Appendix 6 showcases the targeted editing by Qwen-Image-Edit.

## 3.2 EVALUATION AGENT ($A_{\text{EVAL}}$)

$A_{\text{EVAL}}$ assesses the quality of each generated image and outputs a structured evaluation report. The assessment is based on two primary metrics: **Aesthetic Quality** and **Text–Image Alignment**, each scored on a 0–10 scale. The prompt used for $A_{\text{EVAL}}$ is in Appendix I.2.

Aesthetic Quality reflects the overall merit of the image, considering composition, color harmony, lighting, focus, sharpness, emotional impact, and uniqueness. Text–Image Alignment measures consistency with the prompt by verifying subject presence, spatial correctness, adherence to style, and background consistency. The overall performance score is computed as the average of the two scores. Beyond scoring, the agent also detects visual artifacts (e.g., noise, tiling, blending errors), identifies missing elements, and provides targeted suggestions for refinement.

If the score falls below a pre-defined threshold, $S_{\text{GEN}}$ triggers an editing loop in which the image is edited via Qwen-Image-Edit using the suggested improvements by generating a refined prompt utilising $A_{\text{ORC}}$ based on suggestions generated by $A_{\text{EVAL}}$. This process repeats until the image reaches a satisfactory score or the maximum editing attempts are exhausted.

## 3.3 GUIDANCE AGENT ($A_{\text{GUID}}$)

The Guidance Agent enables the self-improving component of our workflow. A trajectory denotes a complete generation sequence, from the initial prompt to the final image, while decision nodes mark points where sub-agents perform key actions. $A_{\text{GUID}}$ records each trajectory in a structured format by condensing trajectories into pitfalls and successes for every decision node. These records accumulate in a persistent knowledge base, allowing the system to detect recurring patterns, such as

specific sub-agents consistently failing on certain prompt types. When processing new inputs, $A_{\text{GUID}}$ retrieves relevant trajectories and provides both corrective guidance (derived from past pitfalls and successes) and workflow guidance (a structured description of the process) to each sub-agent. Our hypothesis is that this combination equips sub-agents with richer context and leads to more reliable decisions. The prompts used for $A_{\text{GUID}}$ are in Appendix I.3

**1. Knowledge Base Construction:** The process begins by populating a 'global memory', which serves as the system's persistent knowledge base. Each complete workflow execution, or 'trajectory', is recorded for analysis. Instead of storing raw logs, a post-processing step occurs to analyze each trajectory. At each 'decision node', it extracts structured insights by identifying specific 'pitfalls' and 'successes'. This compressed information is then stored in the knowledge base, which is structured with dedicated databases for different models (Qwen-Image and Qwen-Image-Edit) to facilitate model-specific pattern recognition.

**2. Guidance Formulation:** When a new user prompt is received, the Guidance Agent uses it as a query to retrieve the $K$-most semantically similar trajectories from the knowledge base via RAG. The agent then analyzes the structured data from these retrieved trajectories, aggregating the recorded pitfalls and successes associated with each decision node. By identifying high-frequency patterns such as a sub-agent repeatedly failing on a certain kind of prompt, it synthesizes a corrective guidance. This guidance contains actionable instructions, such as specific negative prompts to add or adjustments to be made, designed to mitigate known failure modes and replicate successful outcomes for the current task. Additionally, it also synthesizes a static workflow guidance, which provides each decision node about the overall workflow and its role in it.

**3. Guidance Injection.** The synthesized guidance is then injected into the active workflow. At each decision node, the corresponding sub-agent receives both the static workflow guidance and the dynamic corrective guidance (formulated in the previous step) as part of its input context. The sub-agent then acts on this information. The static workflow guidance helps each decision node anticipate how its result not only affects the generation output but also the performance of other sub-agents. For instance, if the model is prompted to generate an image of a wall clock, ($A_{\text{GUID}}$) learns that ($S_{\text{GEN}}$) is unable to generate wall clocks that show a time other than 10:10 (Pawar, 2025; Harris, 2025), and hence it would suggest ($A_{\text{ORC}}$) to explicitly guide the generation towards having the wall clock showing 10:10 and an overall low creativity.

The operation of $A_{\text{GUID}}$ can be understood in three stages: knowledge base construction, guidance formulation, and guidance injection. This cyclical process of retrieving guidance, applying it, and learning from the outcome establishes a robust, training-free feedback loop. It is through this iterative refinement that the system continuously enhances its performance and adapts its strategies based on accumulated experience.

## 4 EXPERIMENTS

### 4.1 EXPERIMENTATION SETUP

All experiments are conducted on a single node equipped with $8 \times 80$GB NVIDIA A100 GPUs. The Qwen-2.5-72B-VL[1] model is hosted on 4 GPUs using vLLM (Kwon et al., 2023), while the remaining 4 GPUs are allocated for image generation. The auxiliary models Qwen-Image (Wu et al., 2025a), Qwen-Edit (Wu et al., 2025a), and Qwen-Embedding (Zhang et al., 2025) together require approximately 47GB of VRAM on a single GPU, with Qwen-Image and Qwen-Edit deployed under 4-bit quantization to reduce memory footprint and improve the speed. We built our multi-agent framework using LangGraph (team, 2024). During the image generation process, if a generated image received a score below 8.0 from $A_{\text{EVAL}}$, the system triggered editing, with a maximum of two attempts, following T2I-Copilot (Chen et al., 2025a).

The guidance system was implemented with a SQLite database storing condensed trajectories. Once 200 trajectories were accumulated, the database was used for Retrieval-Augmented Generation (RAG) (Lewis et al., 2020). For each new prompt, we retrieved the top-$k = 5$ most semantically similar trajectories using Qwen-Embedding-0.6B with FAISS (Douze et al., 2024).

---

[1]https://huggingface.co/Qwen/Qwen2.5-VL-72B-Instruct

| Method | GenAI-Bench | | | | | | | | | | | | | DrawBench |
|---|---|---|---|---|---|---|---|---|---|---|---|---|---|---|
| | Basic | | | | | | Advanced | | | | | | Overall | |
| | Attribute | Scene | Relation | | | Overall | Count | Differ | Compare | Logical | | Overall | | |
| | | | Spatial | Action | Part | | | | | Negate | Universal | | | |
| *Proprietary* | | | | | | | | | | | | | | |
| Imagen 3 v002 (Baldridge et al., 2024b) | 0.909 | 0.923 | 0.909 | 0.903 | 0.918 | 0.912 | 0.841 | 0.841 | 0.795 | 0.673 | 0.788 | 0.776 | 0.839 | 0.866 |
| (Task completion rate) | (92.4%) | (93.1%) | (93.0%) | (89.3%) | (88.7%) | (92.3%) | (91.5%) | (88.9%) | (90.7%) | (88.8%) | (89.1%) | (90.7%) | (91.4%) | - |
| - | (97.0%) | | | | | | | | | | | | | |
| Recraft v3 (Recraft, 2024) | 0.914 | 0.913 | 0.913 | 0.901 | 0.913 | 0.913 | 0.806 | 0.797 | 0.772 | 0.589 | 0.761 | 0.725 | 0.811 | 0.836 |
| FLUX1.1-pro (Labs, 2024) | 0.890 | 0.899 | 0.884 | 0.871 | 0.894 | 0.884 | 0.766 | 0.788 | 0.751 | 0.490 | 0.710 | 0.666 | 0.766 | 0.786 |
| Midjourney v6 (Midjourney, 2024) | 0.880 | 0.870 | 0.870 | 0.870 | 0.910 | 0.870 | 0.780 | 0.780 | 0.790 | 0.500 | 0.760 | 0.690 | 0.772 | - |
| DALL-E 3 (OpenAI, 2024) | 0.910 | 0.900 | 0.920 | 0.890 | 0.910 | 0.900 | 0.820 | 0.780 | 0.820 | 0.480 | 0.800 | 0.700 | 0.791 | - |
| *Open-source* | | | | | | | | | | | | | | |
| Kolors v1.0 (Team, 2024) | 0.821 | 0.841 | 0.832 | 0.818 | 0.803 | 0.819 | 0.737 | 0.726 | 0.705 | 0.438 | 0.695 | 0.621 | 0.711 | 0.646 |
| Playground v2.5 (Li et al., 2024b) | 0.818 | 0.850 | 0.803 | 0.818 | 0.821 | 0.815 | 0.732 | 0.696 | 0.721 | 0.499 | 0.695 | 0.640 | 0.720 | 0.743 |
| HunyuanDiT v1.2 (Li et al., 2024c) | 0.817 | 0.855 | 0.825 | 0.827 | 0.798 | 0.818 | 0.732 | 0.723 | 0.743 | 0.475 | 0.692 | 0.640 | 0.721 | 0.712 |
| Janus Pro-7B (Chen et al., 2025c) | 0.865 | 0.886 | 0.867 | 0.856 | 0.870 | 0.859 | 0.731 | 0.759 | 0.734 | 0.480 | 0.693 | 0.653 | 0.747 | 0.786 |
| Lumina-Image-2.0 (Team, 2025) | 0.879 | 0.896 | 0.876 | 0.872 | 0.885 | 0.874 | 0.760 | 0.767 | 0.729 | 0.451 | 0.723 | 0.649 | 0.752 | 0.790 |
| SD 3.5 large (AI, 2024) | 0.891 | 0.895 | 0.889 | 0.880 | 0.895 | 0.890 | 0.760 | 0.763 | 0.743 | 0.471 | 0.707 | 0.659 | 0.764 | 0.781 |
| FLUX.1-dev (Labs, 2024) | 0.873 | 0.875 | 0.862 | 0.853 | 0.875 | 0.864 | 0.747 | 0.756 | 0.733 | 0.456 | 0.711 | 0.646 | 0.745 | 0.769 |
| GenArtist (Wang et al., 2024c) | 0.702 | 0.736 | 0.659 | 0.677 | 0.688 | 0.693 | 0.553 | 0.473 | 0.518 | 0.437 | 0.546 | 0.504 | 0.588 | 0.607 |
| T2I-Copilot (Chen et al., 2025a) | 0.893 | 0.909 | 0.893 | 0.885 | 0.899 | 0.892 | 0.813 | 0.807 | 0.759 | 0.659 | 0.766 | 0.747 | 0.813 | 0.829 |
| *Qwen-Image-Based* | | | | | | | | | | | | | | |
| Qwen-Image | 0.857 | 0.874 | 0.858 | 0.853 | 0.861 | 0.853 | 0.766 | 0.777 | 0.753 | 0.501 | 0.738 | 0.677 | 0.757 | 0.853 |
| Qwen-Agents | 0.906 | 0.912 | 0.910 | 0.897 | 0.921 | 0.907 | 0.790 | 0.814 | 0.777 | 0.497 | 0.742 | 0.690 | 0.789 | 0.833 |
| Qwen-Agent$_{neg}$ | 0.883 | 0.900 | 0.885 | 0.882 | 0.888 | 0.884 | 0.824 | 0.820 | 0.777 | 0.684 | 0.760 | 0.764 | 0.818 | 0.808 |
| Qwen-Agents$_{vneg}$ | 0.913 | 0.912 | 0.913 | 0.905 | 0.929 | 0.913 | 0.841 | 0.838 | 0.795 | 0.700 | 0.783 | 0.782 | 0.842 | 0.849 |
| *SIDiffAgent* | 0.919 | 0.924 | 0.918 | 0.909 | 0.928 | 0.918 | 0.857 | 0.855 | 0.805 | 0.715 | 0.797 | 0.796 | 0.852 | 0.860 |
| *SIDiffAgent$_{ep2}$* | **0.940** | **0.942** | **0.939** | **0.934** | **0.947** | **0.939** | **0.890** | **0.879** | **0.833** | **0.771** | **0.837** | **0.836** | **0.884** | **0.901** |

Table 1: **Quantitative comparison of *SIDiffAgent* against 14 methods on DrawBench (Saharia et al., 2022b) and GenAI-Bench (Li et al., 2024a)**, evaluated using VQAScore (Lin et al., 2024). The results for the baselines are taken from T2I-Copilot (Chen et al., 2025a) and our generations were performed with identical random seeds ensuring fairness in evaluation. The best and second-best results are highlighted in **bold** and underlined, respectively.

We evaluated the agentic system on two benchmarks, GenAI-Bench (Li et al., 2024a) and Draw-Bench (Saharia et al., 2022b), using the generation seeds provided in the dataset. We employed VQAScore (Lin et al., 2024) as our primary evaluation metric for image quality and prompt alignment, which was identified as more human-aligned than CLIPScore (Hessel et al., 2021), PickScore (Kirstain et al., 2023) and ImageReward (Xu et al., 2023) by Imagen3 (Baldridge et al., 2024a). We compare *SIDiffAgent* against proprietary models(Imagen (Baldridge et al., 2024b), Recraft (Recraft, 2024), Flux-pro (Labs, 2024), Midjourney (Midjourney, 2024), and DALL-E (OpenAI, 2024)), open-source models (Kolors (Team, 2024), Playground v2.5 (Li et al., 2024b), Hunyuan DiT (Li et al., 2024c), Janus Pro (Chen et al., 2025c), Lumina-Image 2.0 (Team, 2025), SD 3.5 (AI, 2024), Flux dev (Labs, 2024), GenArtist (Wang et al., 2024c), and Qwen Image (Wu et al., 2025a)), and agentic approaches such as T2I-Copilot (Chen et al., 2025a).

## 4.2 ABLATION STUDIES

To understand the contribution of each component, we conduct five ablation studies:

- **Qwen-Image:** Direct generations from Qwen-Image without agents.

- **Qwen-Agents:** Full agentic workflow without adaptive negative prompts or guidance (comparable to T2I-Copilot Chen et al. (2025a), but using Qwen-Image, Qwen-Image-Edit and Qwen-VL).

- **Qwen-Agent$_{neg}$:** Qwen-Agents with a fixed negative prompt (Appendix I), showcasing the role of using negative prompts.

- **Qwen-Agent$_{vneg}$:** Qwen-Agents with variable negative prompt, showcasing the role of using variable negative prompts.

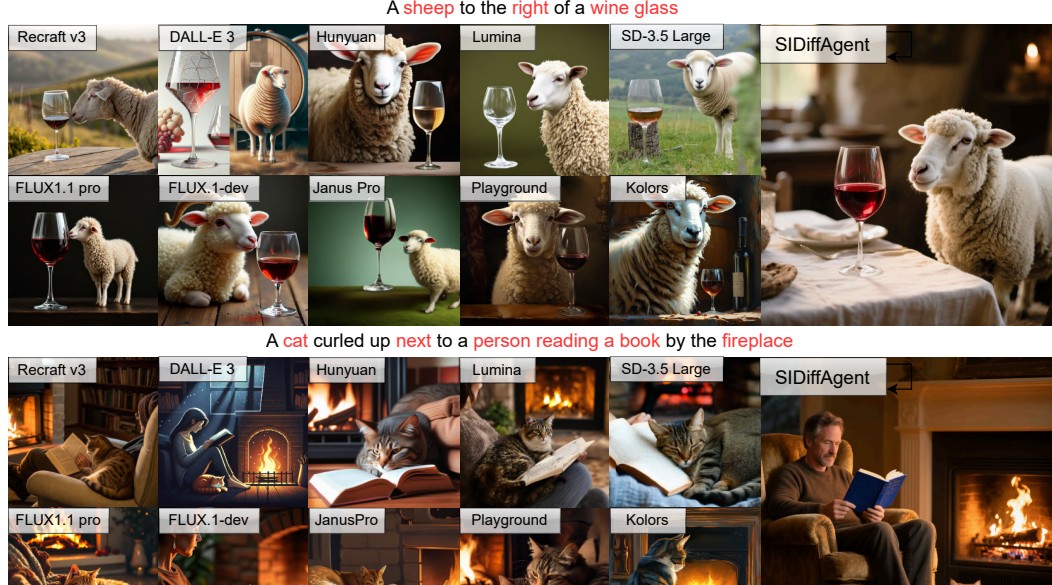

Figure 3: **Qualitative comparison of *SIDiffAgent* with 10 state-of-the-art models on challenging prompts from DrawBench (Top) and GenAI-Bench (Bottom).** *SIDiffAgent* demonstrates superior handling of spatial relationships (correctly placing the sheep 'to the right of' the glass) and compositional complexity (generating all elements of the cat, person, and fireplace scene). In contrast, other models frequently exhibit errors in spatial awareness and object omission.

- ***SIDiffAgent***: Our full framework with adaptive negative prompts and the guidance agent, enabling learning-based improvements from previous trajectories.
- ***SIDiffAgent* Episode 2 (*SIDiffAgent*$_{ep2}$)**: The same setup as above, but leveraging the database accumulated in Episode 1 to assess iterative self-improvement.

## 5 RESULTS

Our experiments demonstrate a clear and progressive improvement at each stage of the proposed framework. Starting with the baseline, Qwen-Image achieves results comparable to leading open-source diffusion models on both GenAI-Bench and DrawBench. Introducing the agentic workflow without guidance(Qwen-Agents) improves performance by +4.22% in VQA Score on GenAI-Bench over the baseline. Incorporating fixed negative prompts (Qwen-Agent$_{neg}$) yields a further improvement of +8.05%, underscoring their importance in refining generation quality. Extending this with adaptive negative prompts, enhanced orchestration, and the guidance agent (*SIDiffAgent*) produces a +12.54% gain over the baseline. Finally, enabling self-improvement through trajectory accumulation in Episode 2 (*SIDiffAgent*$_{ep2}$) leads to the largest gain, achieving a +16.77% improvement over Qwen-Image. Beyond ablations, *SIDiffAgent* surpasses existing state-of-the-art models: it outperforms proprietary systems such as Imagen3 with a +5.39% relative improvement, open-source models such as SD 3.5 with a +15.70% gain, and prior agentic approaches like T2I-Copilot with a +8.73% improvement. Qualitative results in Figure 3 and Appendix K further validate these improvements, with additional qualitative ablation comparisons provided in Figure 9. We additionally conduct a human evaluation study with 50 participants from varied geographical backgrounds to assess subjective preference between our method and the baseline.*SIDiffAgent* is preferred in 69% of cases compared to 31% for T2I-Copilot, demonstrating a clear advantage in perceived visual quality and intent alignment. The inter-annotator agreement, measured using Cohen's $\kappa$, is 0.286, reflecting moderate consistency across raters.

## 6   COST QUALITY TRADEOFF

We conduct a latency analysis between Qwen-image, T2I-Copilot with Qwen, and *SIDiffAgent* on 1024×1024 resolution images using an NVIDIA A6000 GPU, with all VLM models executed identically via OpenRouter to ensure fairness. Although *SIDiffAgent* introduces additional computational

| Method | Inference Time per Prompt (min) |
|---|---|
| Qwen-Image (base) | 0.78 |
| T2I-Copilot (Qwen-Image + Edit) | 1.50 |
| *SIDiffAgent* | 2.31 |

Table 2: End-to-end latency comparison under identical evaluation conditions (1024×1024 resolution, NVIDIA A6000).

overhead per image relative to single-round pipelines, it meaningfully reduces the total user-side cost of achieving a satisfactory final output. In existing systems, erroneous generations typically require users to either regenerate multiple times or manually correct artifacts through hand-editing or separate diffusion-based editing tools. In contrast, *SIDiffAgent* preserves the user's original intent and autonomously executes the full refinement loop, systematically correcting semantic mismatches, visual defects, spatial inconsistencies, and color-related errors without further user intervention.

## 7   GENERALISATION OF *SIDiffAgent*

We additionally evaluate the generalisation of our framework beyond the Qwen family of models, where we use Flux1-dev (Labs, 2024) as the generation model while keeping everything else the same. The results in 3 indicate that the *SIDiffAgent* framework and memory-based self-improvement generalize beyond the Qwen model family.

| Method | DrawBench Avg VQA |
|---|---|
| Flux-dev (base) | 0.7750 |
| Flux-dev-Agent | 0.8217 |
| Flux-dev$_{ep1}$ | 0.8338 |
| Flux-dev$_{ep2}$ | 0.8647 |

Table 3: Effect of agentic refinement and memory episodes on Flux-dev on Drawbench.

## 8   CONCLUSION

In this work, we introduced *SIDiffAgent*, a training-free agentic framework for enhanced text-to-image generation, achieving state-of-the-art results on GenAI-Bench and DrawBench. Our approach refines prompts while preserving the original user intent, incorporates test-time evaluation with a vision-language model for fine-grained edits, and introduces a guidance agent that accumulates knowledge from past trajectories to enable iterative self-improvement. This combination yields substantial gains over both proprietary and open-source state-of-the-art systems, as well as prior agentic frameworks. By systematically integrating prompt engineering, evaluation, editing, and adaptive guidance, *SIDiffAgent* demonstrates that test-time agentic coordination can significantly improve alignment, controllability, and generation quality without additional training. These findings highlight the broader potential of agentic frameworks and prompt engineering for scalable deployment of diffusion-based systems in real-world creative and professional applications.

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

# Appendix

## A  LIMITATIONS

Due to resource constraints, we do not benchmark our model against proprietary state-of-the-art systems such as NanoBanana (team, 2025), GPT-Image (team, 2025), and Flux-Kontext-Pro (Labs et al., 2025). Furthermore, we do not conduct a human evaluation of the generated images, which would provide additional insights into perceptual quality and usability. Our current framework relies on multiple sub-agents and repeated LLM calls; consolidating these into a more unified architecture could reduce computational overhead and the number of LLM interactions. In addition, the Guidance Agent stores prompts and trajectories in its agentic memory. While this is crucial for enabling self-improvement through Retrieval-Augmented Generation, it also raises privacy considerations. Stored prompts may contain personal or sensitive user information.

## B  EVALUATION OF RETRIEVAL QUALITY

To evaluate retrieval quality, we perform an LLM-as-a-judge study using GPT-5-mini on the memory obtained from the first run of GenAI-Bench for epoch 2. Specifically, we compute two metrics: (1) *Overall Score*, which represents the rating when all top-k retrieved trajectories are shown together to the judge, and (2) *Mean Average Score*, which computes the mean score across each retrieved trajectory when rated independently. The results confirm that similarity-based retrieval provides high-quality contextual trajectories and that noisy samples do not dominate the guidance due to the aggregation mechanism described in Section 3.3. Furthermore, performance remains robust across different values of k. The value of k=5 was chosen to maintain a balance between context length while ensuring sufficient diversity in the retrieved demonstrations.

The results for the retrieval quality evaluation are presented in Table 4. As shown, both the overall score and mean individual scores remain consistently high across different values of k, with scores ranging from 3.33 to 3.58 on a 4-point scale. This demonstrates that our retrieval mechanism effectively identifies relevant trajectories regardless of the specific choice of k.

| Top-$k$ | Overall Score (Mean) | Individual Avg Score (Mean) |
|---------|----------------------|------------------------------|
| 3       | 3.50                 | 3.56                         |
| 5       | 3.52                 | 3.46                         |
| 7       | 3.53                 | 3.38                         |
| 10      | 3.58                 | 3.33                         |

Table 4: Retrieval quality evaluation using LLM-as-a-judge.

Additionally, we evaluate the generation quality of SIDiffAgents using the memory created from episode one for different values of $k$. The results, shown in Table 5, indicate that performance remains stable across different retrieval sizes, with average VQA scores ranging from 0.883 to 0.901. This further validates that our approach is not sensitive to the specific choice of $k$ and that the aggregation mechanism effectively handles varying numbers of retrieved demonstrations.

| $k$ | Average VQA Score |
|-----|-------------------|
| 3   | 0.889             |
| 5   | 0.901             |
| 7   | 0.884             |
| 10  | 0.883             |

Table 5: Average VQA scores for different $k$ using memory from episode one.

The complete judge prompt and additional implementation details are provided in the Appendix I

## C  GENERALIZATION OF MEMORY

We test memory generalization when the memory is built exclusively from GenAI-Bench trajectories and evaluated on the unseen DrawBench prompts. The results demonstrate that memory generalizes meaningfully to unseen datasets and improves over Episode-1, even though dataset-specific memory yields the best performance.

| Experimental Setting | Performance Score |
|---|---|
| Episode-1 (No memory) | 0.860 |
| GenAI-memory (DrawBench Generalization) | 0.8725 |
| DrawBench-native memory | 0.901 |

Table 6: Performance comparison across different memory settings.

## D  ADDITIONAL BASELINE COMPARISON

We additionally compare prior to plug-and-play and multi-round prompting systems to *SIDiffAgent*. We conduct experiments on LLM-Grounded Diffusion (LLM-D) Lian et al. (2023) and Self-Correcting LLM-Controlled Diffusion (SLD) Wu et al. (2024a) on DrawBench. The results are summarized in Table 7. As shown, both LLM-D (0.5317) and SLD (0.6326) perform far below Qwen-Image (0.853) and significantly below SIDiffAgent (0.860 in Episode 1 and 0.901 in Episode 2).

| Method | VQA Score |
|---|---|
| LLM-D [1] | 0.5317 |
| SLD [2] | 0.6326 |
| Qwen-Image (base) | 0.8530 |
| SIDiffAgent (Episode 1) | 0.8600 |
| SIDiffAgent (Episode 2) | 0.9010 |

Table 7: Comparison of VQA Scores under the DrawBench evaluation protocol. Earlier LLM-based prompting pipelines underperform relative to SIDiffAgent.

## E  FAILURE CASES OF *SIDiffAgent*

We observe that the multi-agent framework may lead to a decrease in the generated images, consistent with the failure of multi-agent systems as seen in previous work Cemri et al. (2025).This occurs particularly:

- When the memory contains similar prompts with conflicting outcomes, the guidance can occasionally mislead agents. Some examples in  4
- For complex generation tasks, iterative regeneration may enhance certain features but inadvertently worsen others, preventing consistent improvement across iterations. Example of this in Table  5 first set of regeneration.
- When the creativity agent proposes rare or compositional attributes that the generator systematically fails to render, the multi-agent system may enter unnecessary correction loops. Example of this in Table  5 second set of regeneration.

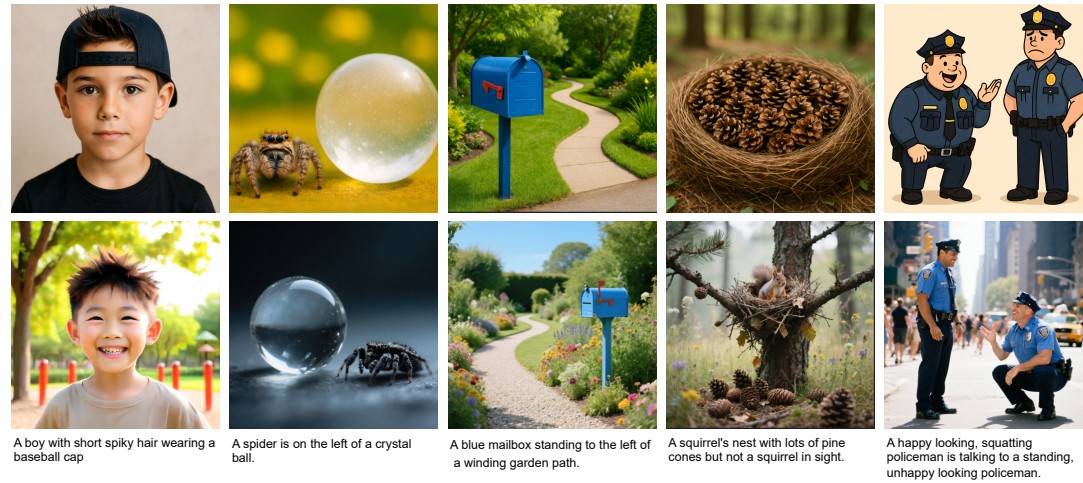

A boy with short spiky hair wearing a baseball cap

A spider is on the left of a crystal ball.

A blue mailbox standing to the left of a winding garden path.

A squirrel's nest with lots of pine cones but not a squirrel in sight.

A happy looking, squatting policeman is talking to a standing, unhappy looking policeman.

Figure 4: The first row shows images generated by T2I-Copilot without memory, while the second row shows outputs from our approach. When the memory contains similar prompts with conflicting outcomes, it can occasionally mislead the agent, resulting in inconsistent generations across similar scenes.

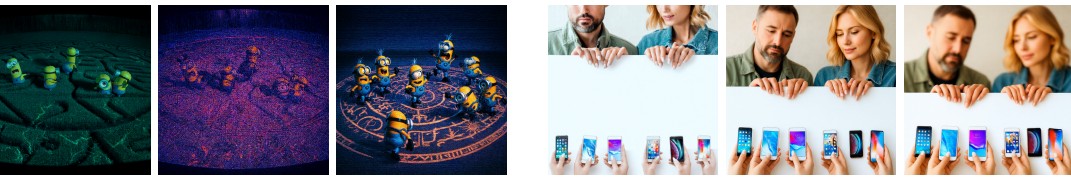

a distorted, low-resolution video feed from a trail camera, displaying a bizarre scene of animated minion figurines moving erratically within a crop circle that appears cursed and sinister. the image is plagued with digital artifacts, creating a grainy texture that adds to the unsettling atmosphere. flashes of datamoshing cause the figures to appear fused and deformed, creating a chilling effect as they twist and contort in unnatural ways.

Eight white hands, all gripped around the edges of sleek smartphones, are positioned at the bottom of the frame. The phones display colorful screens visible against the backdrop of a light-textured wall. Above two of the hands, the lower halves of faces are in view; one features the bottom of a male's face with an unshaven chin, likely in his mid-thirties, wearing a light green canvas jacket, and the other, a female with sun-kissed blond hair, is dressed in a classic blue denim jacket. Both appear to be engrossed in the content on their devices.

Figure 5: Comparison between initially generated and regenerated images. **Left**: In complex scenes with intricate textures and motion-like artifacts, iterative regeneration amplifies some visual details while degrading others, leading to unstable convergence across iterations. **Right**: When the creativity agent suggests uncommon or compositionally difficult attributes, the generator repeatedly fails to realize them, triggering unnecessary correction loops and producing minimal meaningful improvement over successive regenerations.

## F  ADDITIONAL QUANTITATIVE RESULTS

Additional quantitative results are shown in Table 8 and Table 9.

Table 8: Quantitative evaluation results on DPG (Hu et al., 2024).

| Model | Global | Entity | Attribute | Relation | Other | Overall↑ |
|---|---|---|---|---|---|---|
| SD v1.5 (Rombach et al., 2021) | 74.63 | 74.23 | 75.39 | 73.49 | 67.81 | 63.18 |
| PixArt-$\alpha$ (Chen et al., 2024b) | 74.97 | 79.32 | 78.60 | 82.57 | 76.96 | 71.11 |
| Lumina-Next (Zhuo et al., 2024) | 82.82 | 88.65 | 86.44 | 80.53 | 81.82 | 74.63 |
| SDXL (Podell et al., 2024) | 83.27 | 82.43 | 80.91 | 86.76 | 80.41 | 74.65 |
| Playground v2.5 (Li et al., 2024b) | 83.06 | 82.59 | 81.20 | 84.08 | 83.50 | 75.47 |
| Hunyuan-DiT (Li et al., 2024c) | 84.59 | 80.59 | 88.01 | 74.36 | 86.41 | 78.87 |
| Janus (Wu et al., 2025b) | 82.33 | 87.38 | 87.70 | 85.46 | 86.41 | 79.68 |
| PixArt-$\Sigma$ (Chen et al., 2024a) | 86.89 | 82.89 | 88.94 | 86.59 | 87.68 | 80.54 |
| Emu3-Gen (Wang et al., 2024b) | 85.21 | 86.68 | 86.84 | 90.22 | 83.15 | 80.60 |
| Janus-Pro-1B (Chen et al., 2025b) | 87.58 | 88.63 | 88.17 | 88.98 | 88.30 | 82.63 |
| DALL-E 3 (OpenAI, 2024) | 90.97 | 89.61 | 88.39 | 90.58 | 89.83 | 83.50 |
| FLUX.1 [Dev] (Labs, 2024) | 74.35 | 90.00 | 88.96 | 90.87 | 88.33 | 83.84 |
| SD3 Medium (Esser et al., 2024) | 87.90 | 91.01 | 88.83 | 80.70 | 88.68 | 84.08 |
| Janus-Pro-7B (Chen et al., 2025c) | 86.90 | 88.90 | 89.40 | 89.32 | 89.48 | 84.19 |
| HiDream-I1-Full (Cai et al., 2025) | 76.44 | 90.22 | 89.48 | 93.74 | 91.83 | 85.89 |
| Lumina-Image 2.0 (Team, 2025) | - | 91.97 | 90.20 | **94.85** | - | 87.20 |
| Seedream 3.0 (Gao et al., 2025) | 94.31 | **92.65** | 91.36 | 92.78 | 88.24 | 88.27 |
| GPT Image 1 [High] (OpenAI, 2025) | 88.89 | 88.94 | 89.84 | 92.63 | 90.96 | 85.15 |
| Qwen-Image (Wu et al., 2025a) | 91.19 | 92.58 | 92.61 | 87.54 | 85.20 | 87.84 |
| Qwen-Agents | 90.58 | 94.54 | 92.93 | 89.48 | 86.40 | 89.51 |
| *SIDiffAgent* | 90.58 | 96.77 | 96.11 | 92.07 | 93.60 | 93.68 |
| *SIDiffAgent*$_{ep2}$ | 93.92 | 97.93 | 97.23 | 93.97 | 95.20 | 95.70 |

Table 9: Quantitative Evaluation results on GenEval (Ghosh et al., 2023).

| Model | Single Object | Two Object | Counting | Colors | Position | Attribute Binding | Overall↑ |
|---|---|---|---|---|---|---|---|
| Show-o (?) | 0.95 | 0.52 | 0.49 | 0.82 | 0.11 | 0.28 | 0.53 |
| Emu3-Gen (Wang et al., 2024b) | 0.98 | 0.71 | 0.34 | 0.81 | 0.17 | 0.21 | 0.54 |
| PixArt-$\alpha$ (Chen et al., 2024b) | 0.98 | 0.50 | 0.44 | 0.80 | 0.08 | 0.07 | 0.48 |
| SD3 Medium (Esser et al., 2024) | 0.98 | 0.74 | 0.63 | 0.67 | 0.34 | 0.36 | 0.62 |
| FLUX.1 [Dev] (Labs, 2024) | 0.98 | 0.81 | 0.74 | 0.79 | 0.22 | 0.45 | 0.66 |
| SD3.5 Large (Esser et al., 2024) | 0.98 | 0.89 | 0.73 | 0.83 | 0.34 | 0.47 | 0.71 |
| JanusFlow (Ma et al., 2025) | 0.97 | 0.59 | 0.45 | 0.83 | 0.53 | 0.42 | 0.63 |
| Lumina-Image 2.0 (Team, 2025) | - | 0.87 | 0.67 | - | - | 0.62 | 0.73 |
| Janus-Pro-7B (Chen et al., 2025b) | 0.99 | 0.89 | 0.59 | 0.90 | 0.79 | 0.66 | 0.80 |
| HiDream-I1-Full (Cai et al., 2025) | 1.00 | 0.98 | 0.79 | 0.91 | 0.60 | 0.72 | 0.83 |
| GPT Image 1 [High] (OpenAI, 2025) | 0.99 | 0.92 | 0.85 | 0.92 | 0.75 | 0.61 | 0.84 |
| Seedream 3.0 (Gao et al., 2025) | 0.99 | 0.96 | 0.91 | 0.93 | 0.47 | 0.80 | 0.84 |
| Qwen-Image | 0.96 | 0.92 | 0.83 | 0.88 | 0.71 | 0.91 | 0.87 |
| Qwen-Agents | 0.96 | 0.95 | 0.84 | 0.96 | 0.70 | 0.89 | 0.88 |
| *SIDiffAgent* | 0.99 | 0.97 | 0.80 | 0.99 | 0.71 | 0.92 | 0.89 |
| *SIDiffAgent*$_{ep2}$ | 0.99 | 0.98 | 0.93 | 1.00 | 0.81 | 0.94 | 0.94 |

# G IMPLEMENTATION DETAILS

Additional implementation details regarding the Guidance Agent ($A_{\text{GUID}}$) are listed below:

**Agentic Memory** The agentic memory implements a hybrid storage approach:

1. **SQLite Schema (Qwen-Image):** Qwen-Image

```
id INTEGER PRIMARY KEY AUTOINCREMENT,
timestamp TEXT NOT NULL,
image_index TEXT,
original_prompt TEXT,
refined_prompt TEXT,
evaluation_score REAL,
confidence_score REAL,
regeneration_count INTEGER,
```

```
            trajectory_reasoning TEXT,
            step_scores TEXT,
            successes TEXT,
            pitfalls TEXT,
            overall_rating REAL,
            config_data TEXT,
            process_summary TEXT
```

2. **SQLite Schema (Qwen-Image-Edit):** Qwen-Image

```
            id INTEGER PRIMARY KEY AUTOINCREMENT,
            timestamp TEXT NOT NULL,
            image_index TEXT,
            original_prompt TEXT,
            refined_prompt TEXT,
            evaluation_score REAL,
            confidence_score REAL,
            regeneration_count INTEGER,
            reference_image TEXT,
            trajectory_reasoning TEXT,
            step_scores TEXT,
            successes TEXT,
            pitfalls TEXT,
            overall_rating REAL,
            config_data TEXT,
            process_summary TEXT
```

3. **Vector Database:**

   - Index Type: FAISS IndexFlatIP
   - Embedding: Qwen-Embedding
   - Search: Approximate Nearest Neighbors

## H  HYPERPARAMETER SETTINGS

Key configuration parameters used in our experiments:

- **Quality Assessment by $A_{\text{EVAL}}$:**
  - Base threshold: 8.0
- **Memory System:**
  - Guidance extraction threshold: 200 samples
  - Similarity search k: 5
- **Generation Sub-Agent ($S_{\text{GEN}}$):**
  - Maximum edits$\leq 2$
  - Guidance scale for Qwen-Image/Qwen-Image-Edit: 4.0
  - Negative prompt weight for Qwen-Image/Qwen-Image-Edit: 1.0

## I  PROMPTS

**Retrieval Group Evaluation Score**

```
You are an expert evaluator assessing the quality of retrieved prompts
    from a text-to-image generation database.

*Query Prompt:*
"{query}"

*Retrieved Prompts (Group):*
{retrieved_text}

*Task:*
```

```
Evaluate the quality of these retrieved prompts *as a group*. Consider
    the following criteria:
1. *Semantic Similarity* -- How directly the group matches the concepts,
    objects, attributes, or themes in the query.
2. *Structural Alignment* -- Even when not directly similar, do the
    prompts contain elements, styles, moods, or conceptual directions
    that could still help guide the generation of images related to the
    query?
3. *Relevance for Guidance* -- How helpful the group collectively would
    be in steering a text-to-image system toward producing an image
    aligned with the query.
4. *Concept Coverage* -- Whether different relevant aspects are
    represented (not redundant).
5. *Overall Alignment* -- Whether the group preserves the core visual
    intent or meaning of the query, including soft/indirect relationships
    .

*Scoring Guidelines:*
- *5*: Excellent -- Strong semantic similarity and/or rich soft alignment
    . Highly useful for guiding generation.
- *4*: Good -- Mostly relevant, with some helpful soft alignment.
- *3*: Acceptable -- Mixed relevance; may rely more on soft connections
    than direct ones.
- *2*: Poor -- Weak similarity, minimal useful soft alignment.
- *1*: Very Poor -- No meaningful direct or soft alignment.

Respond with ONLY a JSON object in the exact format:
{
    "overall_score": <integer 1-5>,
    "reasoning": "<brief explanation>"
}
```

**Retrieval Individual Evaluation Score**

```
You are an expert evaluator assessing the quality of a single prompt from
    a text-to-image generation database.

*Query Prompt:*
"{query}"

*Retrieved Prompts (Group):*
{retrieved_text}

*Task:*
Evaluate the quality of these retrieved prompts *as a group*. Consider
    the following criteria:
1. *Semantic Similarity* -- How directly the group matches the concepts,
    objects, attributes, or themes in the query.
2. *Structural Alignment* -- Even when not directly similar, do the
    prompts contain elements, styles, moods, or conceptual directions
    that could still help guide the generation of images related to the
    query?
3. *Relevance for Guidance* -- How helpful the group collectively would
    be in steering a text-to-image system toward producing an image
    aligned with the query.
4. *Concept Coverage* -- Whether different relevant aspects are
    represented (not redundant).
5. *Overall Alignment* -- Whether the group preserves the core visual
    intent or meaning of the query, including soft/indirect relationships
    .

*Scoring Guidelines:*
- *5*: Excellent -- Strong semantic similarity and/or rich soft alignment
    . Highly useful for guiding generation.
- *4*: Good -- Mostly relevant, with some helpful soft alignment.
```

```
- *3*: Acceptable -- Mixed relevance; may rely more on soft connections
    than direct ones.
- *2*: Poor -- Weak similarity, minimal useful soft alignment.
- *1*: Very Poor -- No meaningful direct or soft alignment.

Respond with ONLY a JSON object in the exact format:
{
    "overall_score": <integer 1-5>,
    "reasoning": "<brief explanation>"
}
```

**Constant Negative Prompt used in Ablations**

```
The tones are vibrant, overexposed, details are unclear, style, work,
    painting, image, still, overall grayish, worst quality, low quality,
    JPEG compression artifacts, ugly, incomplete, extra fingers, poorly
    drawn hands, poorly drawn faces, deformed, disfigured, distorted
    limbs, merged fingers, motionless image, cluttered background, three
    legs, many people in the background
```

Below are the prompts for each agent and their sub-agents. The actual prompt sent to an agent is formed by concatening the Guidance Prompt with the System Prompt.

## I.1 GENERATION ORCHESTRATOR AGENT

**Creativity Analysis Sub-Agent**

```
You are an expert at analyzing image generation prompts to determine the
    appropriate creativity level.

The PRIMARY RULE for assessing creativity level is: Shorter and less
    informed prompts require HIGH creativity to detailed and well-
    specified prompts require LOW creativity.

Analyze the given prompt and determine the creativity level based on
    these criteria:

HIGH Creativity Level (system should be highly creative and autonomous):
- Very brief or vague prompts with minimal information (e.g., "a black
    cat", "a blue landscape", "a beautiful sunset")
- Single-phrase or short sentence prompts lacking descriptive details
- Abstract concepts or artistic requests with minimal guidance (e.g., "
    surreal dream", "impressionist style")
- Prompts with numerous undefined elements requiring creative decisions
- Prompts that offer minimal context, leaving most details to be
    creatively determined
- Word count typically under 10 words

MEDIUM Creativity Level (balanced approach):
- Prompts with moderate detail but still containing unspecified aspects
- Prompts specifying subject and some context but lacking specific style
    or compositional elements
- Standard scene descriptions that mention key elements but leave
    secondary elements unspecified
- Prompts with a balance of specific instructions and areas requiring
    creative interpretation
- Word count typically between 10-25 words

LOW Creativity Level (stick closely to specifications):
- Highly detailed and comprehensive prompts with explicit requirements
- Technical or precise requests with specific parameters (e.g., "
    professional headshot photo with precise lighting setup")
- Prompts that explicitly specify style, composition, colors, lighting,
    background, and other details
- Professional or commercial image requests with clear technical
    specifications
```

```
- Prompts that leave very little room for creative interpretation
- Word count typically over 25 words with numerous specific details

Return JSON with:
{
    "creativity_level": "LOW|MEDIUM|HIGH",
    "reasoning": "Detailed explanation of why this creativity level was
        chosen",
    "prompt_characteristics": {
        "detail_level": "low|medium|high",
        "specificity": "vague|moderate|precise",
        "artistic_freedom": "constrained|balanced|open"
    }
}

Examples:

Input: "a cat"
Output: {
    "creativity_level": "HIGH",
    "reasoning": "Extremely brief prompt with no details about breed,
        color, pose, setting, lighting, or style. System must autonomously
         determine all visual elements and composition.",
    "prompt_characteristics": {"detail_level": "low", "specificity": "
        vague", "artistic_freedom": "open"}
}

Input: "sunset on the beach"
Output: {
    "creativity_level": "HIGH",
    "reasoning": "Brief prompt that only specifies basic scene elements.
        Requires creative decisions about color palette, composition,
        foreground elements, beach type, mood, and all other visual
        details.",
    "prompt_characteristics": {"detail_level": "low", "specificity": "
        vague", "artistic_freedom": "open"}
}

Input: "A medieval marketplace with people shopping and vendors selling
    goods"
Output: {
    "creativity_level": "MEDIUM",
    "reasoning": "Prompt has clear subject and basic activity but leaves
        many specifics undefined (architecture style, time of day, types
        of goods, clothing styles, weather, atmosphere). Contains 11 words
         with moderate detail level.",
    "prompt_characteristics": {"detail_level": "medium", "specificity": "
        moderate", "artistic_freedom": "balanced"}
}

Input: "Professional headshot of a 30-year-old woman with shoulder-length
     brown hair, wearing a navy blue blazer, neutral beige background,
    studio lighting with soft key light from left side"
Output: {
    "creativity_level": "LOW",
    "reasoning": "Extremely detailed prompt (24 words) with explicit
        specifications for subject, age, hair length, hair color, clothing
        , background color, lighting setup and direction. Almost all
        creative decisions have been predetermined.",
    "prompt_characteristics": {"detail_level": "high", "specificity": "
        precise", "artistic_freedom": "constrained"}
}
```

**Intention Analysis Sub-Agent**

```
1296   You are an expert prompt analyst for image generation. Your role is to
1297       analyze user prompts and extract key elements for generating high-
1298       quality images. You will:
1299
1300   1. Extract Key Elements:
1301   Identify and structure the following aspects of the prompt:
       - Main Subjects: The key objects, characters, or themes present in the
1302          image.
1303   - Attributes: Descriptive traits of subjects (e.g., color, texture,
1304       expression, pose).
1305   - Spatial Relationships: How the subjects are positioned relative to each
          other.
1306   - Background Description: Environment, atmosphere, and additional
1307       contextual elements.
1308   - Composition: Image framing techniques, including: Rule of thirds,
1309       symmetry, leading lines, framing, and balance.
1310   - Color Harmony: Effectiveness of color combinations, contrast, and
1311       saturation.
       - Lighting & Exposure: Brightness, contrast, highlights, and shadows.
1312   - Focus & Sharpness: Depth of field, clarity, and emphasis on subjects.
1313   - Emotional Impact: How well the image conveys emotions or a strong
1314       message.
1315   - Uniqueness & Creativity: Novelty in subject matter, perspective, or
          composition.
1316   - Visual Style: Specific artistic styles, rendering techniques, or
1317       inspirations.
1318   - Reference Images: Directories for content and style reference images.
1319       If reference images are given, incorporate them into the extracted
1320       details. Do not ask the user about reference images unless explicitly
1321        missing.
1322   2. Identify Ambiguities & Missing Information:
1323   Detect elements that need clarification due to:
1324   - Ambiguous terminology: Terms with multiple interpretations requiring
1325       clarification. (e.g., 'apple' could be a fruit or a technology
          company)
1326   - Vague references: Generic terms needing specification (e.g., "a flag"
1327       without stating which country).
1328   - Unspecified visual details: Missing crucial descriptive elements (e.g.,
1329        "a person" without gender, age, or pose).
1330   - Unclear composition or style: Vague artistic direction or missing
1331       technical details.
1332   - Contextual gaps: Information that could significantly affect the image.
1333   - Missing reference images: If reference images are typically expected
          but not provided.
1334
1335   3. Based on the creativity level:
1336   - LOW: Generate specific questions for every unclear element. Creative
1337       fill should be minimal and only for obvious implications.
       - MEDIUM: Fill in common, widely accepted details automatically and ask
1338       for critical clarifications. Creative fill should be conservative and
1339        directly related to the original prompt.
1340   - HIGH: Creatively fill in missing details while maintaining coherence
1341       with the original prompt. Creative fill should enhance, not replace
          or overshadow original elements.
1342
1343   IMPORTANT: Creative fills must always preserve the core intent and
1344       atmosphere of the original prompt. Avoid introducing elements that
1345       change the fundamental nature of the scene.
1346
1347   4. JSON Output Structure
       Return your analysis in the following format:
1348   {
1349       "identified_elements": {
           "main_subjects": [
```

```
            {
                "ENTITY": "ATTRIBUTE",
                "spatial_relationships": ""
            }
        ],
        "background": "",
        "composition": "",
        "color_harmony": "",
        "lighting": "",
        "focus_sharpness": "",
        "emotional_impact": "",
        "uniqueness_creativity": "",
        "visual_style": "",
        "references": {
            "content": [{"REFERENCE_OBJECT_A": "REFERENCE_IMAGE_DIR_A"}],
            "style": "REFERENCE_STYLE_IMAGE_DIR"
        }
    },
    "ambiguous_elements": [
        {
            "element": "",
            "reason": "",
            "suggested_questions": [],
            "creative_fill": ""
        }
    ]
}

### Example 1
Given prompt: "A photo of a person in a red dress"
Creativity level: MEDIUM
{
    "identified_elements": {
        "main_subjects": [
            {
                "person": "red dress",
            }
        ],
        "background": "",
        "composition": "",
        "color_harmony": "",
        "lighting": "",
        "focus_sharpness": "",
        "emotional_impact": "",
        "uniqueness_creativity": "",
        "visual_style": "",
        "references": {
            "content": [],
            "style": ""
        }
    },
    "ambiguous_elements": [
        {
            "element": "person",
            "reason": "Unspecified details such as gender, age, or pose",
            "suggested_questions": [
                "What is the gender of the person?",
                "What age group does the person belong to?",
                "What pose is the person in?"
            ],
            "creative_fill": "Assume a young adult female standing
                confidently"
        },
        {
            "element": "background",
```

```
              "reason": "No background details provided",
              "suggested_questions": [
                "What kind of background do you envision?",
                "Is there a specific setting or location for the photo?"
              ],
              "creative_fill": "A simple, neutral background to highlight the
                  subject"
          }
      ]
}

### Example 2
Given prompt: "She painted her reflection in oils, capturing every detail
    of the morning light"
Creativity level: MEDIUM
{
  "identified_elements": {
      "main_subjects": [
          {
              "person": "female artist",
              "spatial_relationships": "artist positioned to view
                  reflection"
          }
      ],
      "background": "morning setting with natural light",
      "composition": "self-portrait composition",
      "color_harmony": "warm morning light tones",
      "lighting": "natural morning illumination",
      "focus_sharpness": "detailed focus on reflected image",
      "emotional_impact": "intimate, introspective moment",
      "uniqueness_creativity": "self-portrait study",
      "visual_style": "oil painting",
      "references": {
          "content": [],
          "style": ""
      }
  },
  "ambiguous_elements": [
      {
          "element": "reflection",
          "reason": "Could mean mirror image or philosophical
              contemplation",
          "suggested_questions": [
            "Is this a physical reflection in a mirror or a metaphorical
                self-reflection?",
            "If it's a mirror reflection, what type of mirror setup is
                being used?",
            "What perspective is the reflection being painted from?"
          ],
          "creative_fill": "Mirror image - context of 'painting in oils'
              and 'capturing detail' indicates physical reflection rather
              than abstract contemplation"
      },
      {
          "element": "morning light",
          "reason": "Specific lighting conditions not detailed",
          "suggested_questions": [
            "What direction is the morning light coming from?",
            "Are there any specific shadow patterns?",
            "Is it early morning or late morning light?"
          ],
          "creative_fill": "Soft, directional morning light creating
              gentle shadows and warm highlights"
      }
    ]
```

```
1458    }
1459
1460    ### Example 3
1461    Given prompt: "A shiny apple sitting on a desk next to a keyboard"
1462    Creativity level: MEDIUM
1463    {
1464        "identified_elements": {
1465            "main_subjects": [
1466                {
1467                    "apple": "shiny object",
1468                    "keyboard": "computer keyboard",
1469                    "spatial_relationships": "apple positioned next to keyboard
1470                        on desk surface"
1471                }
1472            ],
1473            "background": "desk environment",
1474            "composition": "close-up still life",
1475            "color_harmony": "modern office colors",
1476            "lighting": "clear lighting to show shininess",
1477            "focus_sharpness": "sharp focus on main objects",
1478            "emotional_impact": "clean, modern feel",
1479            "uniqueness_creativity": "juxtaposition of natural/tech elements",
1480            "visual_style": "contemporary photography",
1481            "references": {
1482                "content": [],
1483                "style": ""
1484            }
1485        },
1486        "ambiguous_elements": [
1487            {
1488                "element": "apple",
1489                "reason": "Could refer to either the fruit or an Apple product (
1490                    like an Apple mouse or AirPods)",
1491                "suggested_questions": [
1492                    "Is this referring to the fruit apple or an Apple technology
1493                        product?",
1494                    "If it's a fruit, what variety/color of apple?",
1495                    "If it's an Apple product, which specific device is it?"
1496                ],
1497                "creative_fill": "Red fruit apple - while the desk/keyboard
1498                    setting might suggest tech, without specific tech-related
1499                    context, assume the natural fruit"
1500            },
1501            {
1502                "element": "keyboard",
1503                "reason": "Style and type of keyboard not specified",
1504                "suggested_questions": [
1505                    "What type of keyboard is it (mechanical, membrane, laptop)?"
1506                        ,
1507                    "Is it a specific brand or color of keyboard?",
1508                    "Is it a full-size keyboard or a compact one?"
1509                ],
1510                "creative_fill": "Modern black computer keyboard with white
1511                    backlight"
            }
        ]
    }
```

**Prompt Refinement Sub-Agent**

```
You are a Qwen prompt expert. Your PRIMARY GOAL is to stay faithful to
    the original prompt while resolving ambiguities. CRITICAL: The
    refined prompt must preserve the core intent, subjects, and
    atmosphere of the original prompt.

        GROUNDING PRINCIPLES:
```

```
            - PRESERVE ALL original subjects, objects, and key elements
                mentioned in the original prompt
            - MAINTAIN the original scene's core atmosphere, mood, and
                context
            - ONLY ADD details that directly support or clarify the original
                prompt
            - AVOID introducing new subjects, objects, or concepts not
                clearly implied by the original
            - Creative filling should ENHANCE, not REPLACE or OVERSHADOW
                original elements

            Steps:
            1. START with the original prompt as the foundation - preserve
                its core structure and intent
            2. RESOLVE ambiguous elements using creative_fill from analysis,
                 but only for true ambiguities
            3. ADD minimal supporting details ONLY if creativity_level is
                 MEDIUM or HIGH AND they enhance the original concept
            4. ENSURE the refined prompt feels like a clearer version of the
                 original, not a different scene
            5. If there is reference image, must keep the its directory

            Return a JSON with:
            {
                "refined_prompt": "A refined version that stays closely
                    grounded to the original prompt while resolving necessary
                     ambiguities. The result should read as a natural
                    clarification of the original, maintaining its core
                    essence.",
                "reasoning": "Explain how the refinement preserves the
                    original prompt's intent while addressing ambiguities."
            }
```

**Adaptive Negative Prompt Sub-Agent**

```
You are an expert at generating negative prompts for image generation
    models like Qwen-Image and Qwen-Image-Edit.

A negative prompt specifies what should NOT appear in the generated image
    . It helps avoid common issues like:
- Poor quality artifacts (blurry, distorted, low quality, pixelated)
- Unwanted objects or elements that commonly appear in similar scenes
- Inappropriate content or style mismatches
- Technical issues (watermarks, text overlays, borders)

Guidelines:
1. Keep negative prompts concise but comprehensive
2. Focus on common unwanted elements for the specific scene type
3. Include general quality-related terms
4. Avoid being too restrictive - don't negate core elements of the
    positive prompt
5. Consider the context and style of the positive prompt

Return a JSON with:
{
    "negative_prompt": "comma-separated negative prompt terms",
    "reasoning": "explanation of why these negative elements were chosen"
}

Examples:
- Portrait: "blurry, low quality, distorted face, multiple heads, extra
    limbs, watermark, text"
- Landscape: "people, buildings, text, watermark, low quality, blurry,
    oversaturated"
- Fantasy scene: "modern objects, realistic style, low quality, blurry,
    watermark, text"
```

## I.2 EVALUATION AGENT

You are an expert image judge. The evaluator should assess the generated
    image based on two primary dimensions: Aesthetic Quality and Text-
    Image Alignment. Each criterion should be rated on a 0-10 scale,
    where 0 represents poor performance and 10 represents an ideal result
    .

Mainly focus on the original prompt: {config.prompt_understanding['
    original_prompt']}.

1. Aesthetic Quality (0-10) Evaluate the artistic and visual appeal of
    the generated image using the following factors: – Composition:
    Effectiveness of image framing, balance, rule of thirds, leading
    lines, and visual stability. – Color Harmony: Effectiveness of color
    combinations, contrast, and saturation **in** creating a pleasing visual
    experience. – Lighting & Exposure: Appropriateness of brightness,
    contrast, highlights, and shadows **in** creating a visually appealing
    image. – Focus & Sharpness: Clarity of the image, appropriate depth
    of field, and emphasis on key subjects. – Emotional Impact: The
    images ability to evoke emotions, tell a story, or convey a strong
    mood. – Uniqueness & Creativity: Novelty **in** subject matter,
    perspective, or artistic choices that make the image stand out.

2. Text-Image Alignment (0-10) Evaluate how well the generated image
    adheres to the provided prompt, considering key elements from the
    prompt analysis: – Presence of Main Subjects: Whether all key objects
    , characters, or elements explicitly mentioned **in** the prompt appear
    **in** the image. – Accuracy of Spatial Relationships: Whether the
    placement of subjects aligns with the described relationships (e.g.,
    "a cat sitting on a table" should not have the **cat** under the table).
    – Adherence to Style Requirements: If a specific visual style (e.g.,
    "oil painting," "realistic photography") is mentioned, evaluate
    whether the generated image follows this directive. – Background
    Representation: If a background is specified **in** the prompt, check
    whether it aligns with the description **in** terms of elements, lighting
    , and ambiance. *# Scoring Explanation – Each subcategory score (e.g.,
     Composition, Presence of Main Subjects) should be rated from 0 to
    10, where: – 0-3  Poor or missing implementation of the aspect. – 4-6
      Moderate adherence but with noticeable flaws. – 7-9  Strong
    adherence with minor imperfections. – 10  Perfect execution. – Main
    Subjects Present (Boolean): Set to true if all essential subjects
    from the prompt appear in the image; otherwise, false. – Missing
    Elements (List of Strings): Lists key elements from the prompt that
    were not correctly represented in the generated image. – Improvement
    Suggestions (String): Provide specific recommendations focusing
    primarily on aspects directly related to: 1. The original prompt: {
    config.prompt_understanding['original_prompt']} 2. The user provided
    information: {config.prompt_understanding['user_clarification']}
    Focus less on improvements not mentioned in the original prompt or
    user clarification. – Return the results in JSON format with the
    following structure:*
{{
    "aesthetic_reasoning": str,
    "aesthetic_score": {{
        "Composition": float,
        "Color Harmony": float,
        "Lighting & Exposure": float,
        "Focus & Sharpness": float,
        "Emotional Impact": float,
        "Uniqueness & Creativity": float
    }},
    "alignment_reasoning": str,
    "alignment_score": {{
        "Presence of Main Subjects": float,

```
        "Accuracy of Spatial Relationships": float,
        "Adherence to Style Requirements": float,
        "Background Representation": float
    }},
    "artifacts": {{
        "detected_artifacts": [str],
        "artifact_reasoning": str
    }},
    "main_subjects_present": bool,
    "missing_elements": [str],
    "improvement_suggestions": str,
    "overall_reasoning": str,
}}

### Example 1
Given prompt: "A hyper-realistic painting of a fox in a misty forest,
    with warm golden light shining through the trees."
{{
    "aesthetic_reasoning": "Strong artistic composition and emotional
        impact, but mist and golden light are underrepresented.",
    "aesthetic_score": {{
        "Composition": 8.5,
        "Color Harmony": 9.0,
        "Lighting & Exposure": 8.0,
        "Focus & Sharpness": 7.5,
        "Emotional Impact": 9.5,
        "Uniqueness & Creativity": 8.0
    }},
    "alignment_reasoning": "Fox and forest align well, but mist and
        lighting fall short of prompt description.",
    "alignment_score": {{
        "Presence of Main Subjects": 9.0,
        "Accuracy of Spatial Relationships": 8.0,
        "Adherence to Style Requirements": 7.0,
        "Background Representation": 9.0
    }},
    "artifacts": {{
        "detected_artifacts": ["Minor noise in mist rendering"],
        "artifact_reasoning": "Mist appears pixelated due to blending
            inconsistencies."
    }},
    "main_subjects_present": true,
    "missing_elements": ["Mist not prominent enough", "Golden light too
        subtle"],
    "improvement_suggestions": "Enhance mist and intensify golden light
        for better atmosphere.",
    "overall_reasoning": "Strong aesthetics and alignment but weakened
        atmosphere due to faint mist and lighting.",
}}

### Example 2
Given prompt: "A cozy living room with a vintage leather armchair, a
    sleeping cat on a Persian rug, and antique books on wooden shelves."
{{
    "aesthetic_reasoning": "Visually pleasing composition and colors, but
        emotional depth is reduced by missing cat and rug.",
    "aesthetic_score": {{
        "Composition": 8.0,
        "Color Harmony": 8.5,
        "Lighting & Exposure": 7.5,
        "Focus & Sharpness": 8.0,
        "Emotional Impact": 6.5,
        "Uniqueness & Creativity": 7.0
    }},
```

```
     "alignment_reasoning": "Armchair and shelves present, but cat and rug
         absent, reducing prompt fidelity.",
     "alignment_score": {{
         "Presence of Main Subjects": 2.0,
         "Accuracy of Spatial Relationships": 7.5,
         "Adherence to Style Requirements": 8.0,
         "Background Representation": 8.0
     }},
     "artifacts": {{
         "detected_artifacts": ["Texture tiling on bookshelf"],
         "artifact_reasoning": "Bookshelf wood grain repeats unnaturally,
             indicating AI tiling artifact."
     }},
     "main_subjects_present": false,
     "missing_elements": ["No cat", "No Persian rug", "Armchair lacks
         vintage style"],
     "improvement_suggestions": "Add cat on Persian rug and adjust armchair
          to appear vintage.",
     "overall_reasoning": "Good aesthetics but major alignment issues due
         to missing key subjects.",
}}
```

## I.3 GUIDANCE AGENT

**Trajectory Analysis**

```
You are an expert AI model performance analyst. Analyze the {model_name}
    model's performance in this image generation task.

WORKFLOW EXECUTION EXPLANATION:
{process_summary}

EXECUTION-SPECIFIC DATA (in workflow sequence):

CREATIVITY LEVEL SETTING:
 Level: {config_context['prompt_details']['creativity_level']}
 Impact: Determined how autonomously the system handled ambiguous prompt
    elements

INTENTION ANALYSIS RESULTS:
 Original prompt: "{config_context['prompt_details']['original']}"
 Analysis findings: System identified visual elements and ambiguous
    aspects requiring interpretation

PROMPT REFINEMENT OUTPUT:
 Refined prompt: "{config_context['prompt_details']['refined']}"
 Refinement quality: {'Significant refinement applied' if config_context[
    'prompt_details']['original'] != config_context['prompt_details']['
    refined'] else 'Minimal refinement needed'}

NEGATIVE PROMPT CREATION:
 Negative prompt applied: "{config_context['generation_params']['
    negative_prompt']}"
 Purpose: Targeted prevention of unwanted artifacts and quality issues

GENERATION EXECUTION:
 Selected model: {config_context['model_selection']['chosen_model']}
 Selection reasoning: {config_context['model_selection']['
    selection_reasoning']}
 System confidence: {config_context['model_selection']['confidence_score'
    ]}/10
 Reference image used: {'Yes' if config_context['generation_params']['
    has_reference_image'] else 'No'}
 Generation seed: {config_context['generation_params']['seed']}
```

```
EVALUATION RESULTS:
 Automated score: {config_context['evaluation_metrics']['evaluation_score
    ']}/10
 User feedback: {config_context['evaluation_metrics']['user_feedback'] or
    'None provided'}

REGENERATION STATUS:
 Current attempt: #{config_context['system_context']['current_attempt']}
    of {config_context['system_context']['total_attempts']} total
 Improvement suggestions: {config_context['evaluation_metrics']['
    improvement_suggestions'] or 'None from previous cycles'}
 Human oversight: {'Enabled' if config_context['system_context']['
    human_in_loop'] else 'Autonomous operation'}

ANALYSIS REQUIREMENTS:
Analyze this model's performance considering the execution flow:
1. How well did the model respond to the creativity level and prompt
    refinement quality?
2. Effectiveness of negative prompt in preventing unwanted artifacts
3. Quality of prompt polishing for this specific model's characteristics
4. Model selection appropriateness based on the refined prompt
    requirements
5. Technical execution quality visible in the generated image
6. Prompt adherence and creative interpretation balance
7. Reference image utilization (if applicable)

IMPORTANT: You will see the actual generated image(s) below. Provide
    specific visual analysis referencing the execution context.

Return a JSON response with detailed breakdown by workflow trajectory:
{{
   "trajectory_reasoning": "Overall analysis of how the workflow
       execution played out from start to finish, including key decision
       points, transitions between steps, and how each step influenced
       the next. Analyze the logical flow and coherence of the entire
       process.",
   "step_scores": {{
      "creativity_level": "Score from 1-10 how appropriate the creativity
          level setting was for this specific prompt",
      "intention_analysis": "Score from 1-10 how effective the intention
          analysis was for this prompt",
      "prompt_refinement": "Score from 1-10 how well the prompt was
          refined for optimal generation",
      "negative_prompt": "Score from 1-10 how effective the negative
          prompt was in preventing issues",
      "generation": "Score from 1-10 the overall quality of the generated
          image",
      "evaluation": "Score from 1-10 how accurate the system's evaluation
          was"
   }},
   "successes": {{
      "creativity_level": "How well the creativity level setting worked
          for this prompt and model",
      "intention_analysis": "Effectiveness of the intention analysis in
          guiding the process",
      "prompt_refinement": "Quality and appropriateness of the prompt
          refinement",
      "negative_prompt": "How well the negative prompt prevented unwanted
          artifacts",
      "generation": "Model selection appropriateness and generation
          quality",
      "evaluation": "Accuracy of evaluation scoring relative to visual
          quality"
   }},
   "pitfalls": {{
```

```
      "creativity_level": "Issues with creativity level setting for this
          prompt type",
      "intention_analysis": "Missed elements or incorrect analysis in
          intention phase",
      "prompt_refinement": "Problems with refined prompt quality or
          completeness",
      "negative_prompt": "Artifacts that negative prompt failed to
          prevent",
      "generation": "Model selection issues or generation quality
          problems",
      "evaluation": "Evaluation inaccuracies or scoring misalignment"
    }},
    "overall_rating": "Rate from 1-10 the overall effectiveness of the
        entire workflow for this specific prompt"
}}

Focus on specific observations from the actual generated image and how
    each workflow step contributed to or detracted from the final result.
    """
```

Note: If regeneration using Qwen-Image-Edit occurs, all the images are also passed to the $A_{\text{GUID}}$ and the final information is added to both the databases of Qwen-Image and Qwen-Image-Edit.

**Guidance generation from similar trajectories for (Qwen Image)**

```
You are an expert at analyzing patterns in image generation workflows to
    provide SPECIFIC, CONCRETE, and ACTIONABLE guidance.

Your specialty is identifying context-specific techniques that work for
    particular types of images, not general best practices.

Focus on content-specific insights like:
- For portrait images: specific lighting techniques, expression guidance,
    compositional elements
- For landscape scenes: time of day effects, weather condition impacts,
    foreground element placement
- For abstract concepts: style reference importance, compositional
    balance techniques, color palette guidance

Avoid general advice like "add more detail" or "be more specific".
Instead provide domain-specific, technical recommendations based on the
    actual examples you analyze.

Based on the analysis of similar prompts, extract CONCRETE and SPECIFIC
    workflow guidance for the new prompt.

SIMILAR PROMPTS DATA:
{similar_data_text}

{workflow_description}

{task_focus}

Return JSON with this EXACT structure:
{{
    "step_analysis": {{

        "creativity_level_determination": {{
            "success_patterns": "SPECIFIC creativity level patterns that
                worked for this type of prompt",
            "failure_patterns": "SPECIFIC creativity level mistakes observed
                 with similar prompts",
            "impact_on_next": "CONCRETE impact on intention analysis quality
                ",
            "preventive_guidance": "DETAILED advice with SPECIFIC examples
                of what to look for",
```

```
            "recommended_score": "Recommend target score (1-10) based on
                similar examples"
        }},
        "intention_analysis": {{
            "success_patterns": "SPECIFIC intention analysis techniques that
                worked for this subject matter",
            "failure_patterns": "SPECIFIC intention analysis pitfalls
                observed with similar prompts",
            "impact_on_next": "CONCRETE impact on prompt refinement",
            "preventive_guidance": "DETAILED advice with SPECIFIC elements
                to identify",
            "recommended_score": "Recommend target score (1-10) based on
                similar examples"
        }},
        "prompt_refinement": {{
            "success_patterns": "SPECIFIC refinement strategies that
                enhanced similar prompts",
            "failure_patterns": "SPECIFIC refinement mistakes observed in
                similar cases",
            "impact_on_next": "CONCRETE impact on negative prompt generation
                ",
            "preventive_guidance": "DETAILED advice with SPECIFIC refinement
                techniques",
            "recommended_score": "Recommend target score (1-10) based on
                similar examples"
        }},
        "negative_model_selection": {{
            "success_patterns": "SPECIFIC negative prompt and model
                selection strategies that were effective for this subject",
            "failure_patterns": "SPECIFIC negative prompt and model
                selection issues observed in similar cases",
            "impact_on_next": "CONCRETE impact on prompt polishing",
            "preventive_guidance": "DETAILED advice with SPECIFIC negative
                prompt terms and model selection criteria",
            "recommended_score": "Recommend target score (1-10) based on
                similar examples"
        }},
        "image_generation": {{
            "success_patterns": "SPECIFIC generation parameters that worked
                for similar content",
            "failure_patterns": "SPECIFIC generation issues observed with
                this prompt type",
            "impact_on_next": "CONCRETE impact on evaluation accuracy",
            "preventive_guidance": "DETAILED advice with SPECIFIC generation
                settings",
            "recommended_score": "Recommend target score (1-10) based on
                similar examples"
        }},
        "quality_evaluation": {{
            "success_patterns": "SPECIFIC evaluation criteria effective for
                this content type",
            "failure_patterns": "SPECIFIC evaluation pitfalls observed with
                similar outputs",
            "impact_on_next": "CONCRETE impact on regeneration decision",
            "preventive_guidance": "DETAILED advice with SPECIFIC quality
                indicators",
            "recommended_score": "Recommend target score (1-10) based on
                similar examples"
        }},
        "regeneration_decision": {{
            "success_patterns": "SPECIFIC decision criteria that led to
                successful outcomes",
            "failure_patterns": "SPECIFIC regeneration mistakes observed
                with similar content",
            "impact_on_next": "N/A - final step",
```

```
            "preventive_guidance": "DETAILED advice with SPECIFIC decision
                factors",
            "recommended_score": "Recommend target score (1-10) based on
                similar examples"
        }}
    }},

    "workflow_insights": {{
        "critical_dependencies": "SPECIFIC step dependencies most relevant
            to this prompt type",
        "common_failure_chains": "CONCRETE failure patterns observed in
            similar cases",
        "success_combinations": "SPECIFIC combinations of choices that
            worked well for this content",
        "overall_rating_prediction": "Predict the likely overall success
            rating (1-10) for this prompt type"
    }}
}}

Your guidance must be SPECIFIC to the prompt type and content domain, not
    generic best practices.
For example, instead of 'Use detailed prompts', say 'For architectural
    images, specify architectural style, materials, lighting conditions,
    and surrounding environment'.
Use concrete, actionable advice derived from the data, not theoretical
    recommendations.

IMPORTANT: Ensure the guidance provides DOMAIN-SPECIFIC advice for the
    content type in the prompts, not just generic image generation tips.
```

**Guidance generation from similar trajectories for (Qwen Image Edit)**

```
You are an expert at analyzing patterns in image generation workflows to
    provide SPECIFIC, CONCRETE, and ACTIONABLE guidance.

Your specialty is identifying context-specific techniques that work for
    particular types of images, not general best practices.

Focus on content-specific insights like:
- For portrait images: specific lighting techniques, expression guidance,
    compositional elements
- For landscape scenes: time of day effects, weather condition impacts,
    foreground element placement
- For abstract concepts: style reference importance, compositional
    balance techniques, color palette guidance

Avoid general advice like "add more detail" or "be more specific".
Instead provide domain-specific, technical recommendations based on the
    actual examples you analyze.

Based on the analysis of similar prompts, extract CONCRETE and SPECIFIC
    workflow guidance for the new prompt.

SIMILAR PROMPTS DATA:
{similar_data_text}

{workflow_description}

{task_focus}

Return JSON with this EXACT structure:
{{
    "step_analysis": {{

        "image_editing": {{
```

```
        "success_patterns": "SPECIFIC editing parameters and techniques
            that worked for similar content and edit types",
        "failure_patterns": "SPECIFIC editing issues observed with this
            edit operation type",
        "impact_on_next": "CONCRETE impact on evaluation accuracy",
        "preventive_guidance": "DETAILED advice with SPECIFIC editing
            settings, reference image usage, and blending techniques",
        "recommended_score": "Recommend target score (1-10) based on
            similar examples"
    }},
    "quality_evaluation": {{
        "success_patterns": "SPECIFIC evaluation criteria effective for
            this edit content type",
        "failure_patterns": "SPECIFIC evaluation pitfalls observed with
            similar edit outputs",
        "impact_on_next": "CONCRETE impact on re-edit decision",
        "preventive_guidance": "DETAILED advice with SPECIFIC edit
            quality indicators and success metrics",
        "recommended_score": "Recommend target score (1-10) based on
            similar examples"
    }},

    "workflow_insights": {{
        "critical_dependencies": "SPECIFIC step dependencies most relevant
            to this prompt type",
        "common_failure_chains": "CONCRETE failure patterns observed in
            similar cases",
        "success_combinations": "SPECIFIC combinations of choices that
            worked well for this content",
        "overall_rating_prediction": "Predict the likely overall success
            rating (1-10) for this prompt type"
    }}
}}

Your guidance must be SPECIFIC to the prompt type and content domain, not
    generic best practices.
For example, instead of 'Use detailed prompts', say 'For architectural
    images, specify architectural style, materials, lighting conditions,
    and surrounding environment'.
Use concrete, actionable advice derived from the data, not theoretical
    recommendations.

IMPORTANT: Ensure the guidance provides DOMAIN-SPECIFIC advice for the
    content type in the prompts, not just generic image generation tips.
```

## J  ALGORITHM

---

**Algorithm 1** Main algorithm for *SIDiffAgent*

---

**Input:** User prompt $P$, Models: QI (Qwen-Image), QIE (Qwen-Image-Edit), QE (Qwen-Embedding), Knowledge base $KB$ (initially may be empty), Hyperparams: $\tau$ (eval threshold), $E$ (max edits), $K$ (retrieval size)

**Output:** Final image $I^*$, final prompt $F$, recorded trajectory $T$

1 **Function** SIDiffAgent($P$)**:**

2    $G \leftarrow$ RetrieveGuidance($KB, P, K$)    // retrieve and generate guidance

3    $c \leftarrow S_{\text{CRE}}(P, G)$                   // creativity level: {low,med,high}

4    $S \leftarrow S_{\text{INT}}(P, c, G)$        // semantic specification, disambiguations

5    $P_{pos} \leftarrow S_{\text{REF}}(P, S, G)$              // refined positive prompt

6    $P_{neg} \leftarrow S_{\text{NEG}}(P, P_{pos}, G)$      // adaptive negative prompt + universal safeguards

7    $edits \leftarrow 0$   $I \leftarrow \varnothing$   $T \leftarrow \varnothing$   **while** $edits \leq E$ **do**

8      $I \leftarrow S_{\text{GEN}}(P_{pos}, P_{neg}, I)$                      // use QI

9      $report \leftarrow A_{\text{EVAL}}(I, P, P_{pos}, G)$      // returns scores and suggestions

10      Record decision node results in $T$ via RecordTrajectory()   $score \leftarrow$ average(report.aesthetic, report.alignment)

11      **if** $score \geq \tau$ **then**

12        **break**                             // satisfactory output

13      **else**

14        $P_{pos}^{new}, P_{neg}^{new} \leftarrow$ re-run $A_{\text{ORC}}$; **if** *report suggests localized correction* **and** *edits* $< E$ **then**

15          $I \leftarrow S_{\text{GEN}}(P_{pos}^{new}, P_{neg}^{new})$, edit, $I$              // use QIE

16          $edits \leftarrow edits + 1$

17      **end**

18    **end**

19    UpdateKnowledgeBase($KB, T$)   **return** $I, P_{pos}, P_{neg}, T$

20 **end**

---

---

**Algorithm 2** Helper functions for main algorithm for *SIDiffAgent*

---

**Function** $S_{CRE}(P, G)$**:**
    analyse guidance G, compute length, specificity cues,
        presence of attributes **if** *very short or vague* **then**
        | **return** *high*
    **else if** *moderately detailed* **then**
        | **return** *medium*
    **else**
        | **return** *low*
    **end**
**end**

**Function** $S_{INT}(P, c, G)$**:**
    analyse guidance G, extract main subjects, attributes,
        relations, composition, lighting, style **if** $c = high$ **then**
        | infer missing attributes using templates and priors
    **else if** $c = medium$ **then**
        | propose optional clarifications and minor assumptions
    **else**
        | preserve original prompt avoid adding assumptions
    **end**
    **return** *structured specification S*
**end**

**Function** $S_{REF}(P, S, G)$**:**
    re-order specification into coherent positive prompt
        text $P_{pos}$ enforce lexical templates for model compatibility **return** $P_{pos}$
**end**

**Function** $S_{NEG}(P, P_{pos}, G)$**:**
    $U \leftarrow$ universal safeguards (e.g., low quality,
        blurry, watermark) derive scene-specific negations from $P_{pos}$ compose concise $P_{neg} \leftarrow U \cup$
    scene-specific
        negations **return** $P_{neg}$
**end**

**Function** $S_{GEN}(P_{pos}, P_{neg}, mode, I_{in} = \varnothing)$**:**
    **if** *mode = initial* **then**
        | **return** *QI.generate(*$P_{pos}, P_{neg}$*)*
    **else if** *mode = edit* **then**
        | **return** *QIE.edit(*$I_{in}, P_{pos}, P_{neg}$*)*
**end**

**Function** $A_{EVAL}(I, P, P_{pos})$**:**
    $a \leftarrow A_{\text{EVAL}}$.scoreAesthetic(I) $\alpha \leftarrow A_{\text{EVAL}}$.scoreAlignment($I, P$ or $P_{pos}$) *sugs* $\leftarrow$
    $A_{\text{EVAL}}$.suggestFixes(I) *arts* $\leftarrow A_{\text{EVAL}}$.detectArtifacts(I) **return** *report with* $(a, \alpha, sugs, arts)$
**end**

**Function** RetrieveGuidance($KB, P, K$)**:**
    **if** $KB$ *empty* **then**
        | **return** $\varnothing$
    embed $P$ into vector $v$ retrieve top-$K$ trajectories
        by similarity aggregate pitfalls and successful actions across
        trajectories **return** *guidance G (structured corrective +*
        *workflow guidance)*
**end**

**Function** RecordTrajectory()**:**
    For each decision node, store: node-id, inputs,
        actions, outputs, scores
**end**

**Function** UpdateKnowledgeBase($KB, T$)**:**
    compress $T$ into node-wise summaries of pitfalls
        and successes append to model-specific sections in $KB$ optionally re-index embeddings for
    efficient retrieval
**end**

---

## K  QUALITATIVE RESULTS

**Impact of Creativity Level**

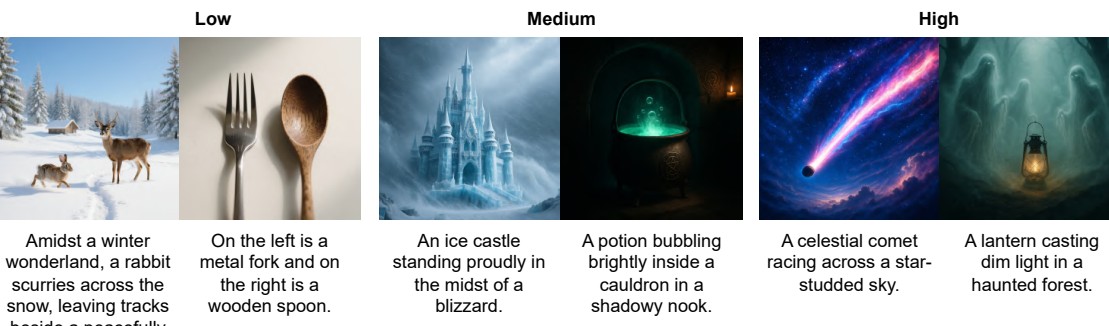

Mountains with a lake (simple prompt)

A hiker standing on a cliff looking at a valley where a river splits into two streams around a small island (Complex Prompt)

**Prompts and generations across creativity levels (set by Creativity Analysis Subagent)**

Amidst a winter wonderland, a rabbit scurries across the snow, leaving tracks beside a peacefully standing deer.

On the left is a metal fork and on the right is a wooden spoon.

An ice castle standing proudly in the midst of a blizzard.

A potion bubbling brightly inside a cauldron in a shadowy nook.

A celestial comet racing across a star-studded sky.

A lantern casting dim light in a haunted forest.

**Some example of edits performed by Qwen-Edit**

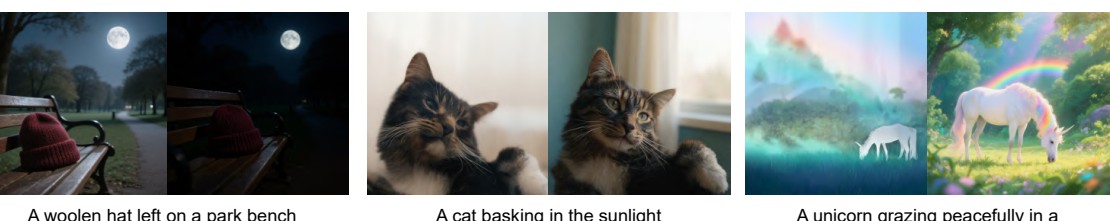

A woolen hat left on a park bench under the moonlight.

A cat basking in the sunlight by a window.

A unicorn grazing peacefully in a radiant, rainbow-lit glade

Figure 6: **Top:** Effect of varying creativity levels on simple and complex prompts. **Middle:** Image generations from diverse prompts, with creativity levels determined by $S_{\text{CRE}}$. **Bottom:** Refinements and enhancements performed by Qwen-Edit on initial suboptimal generations.

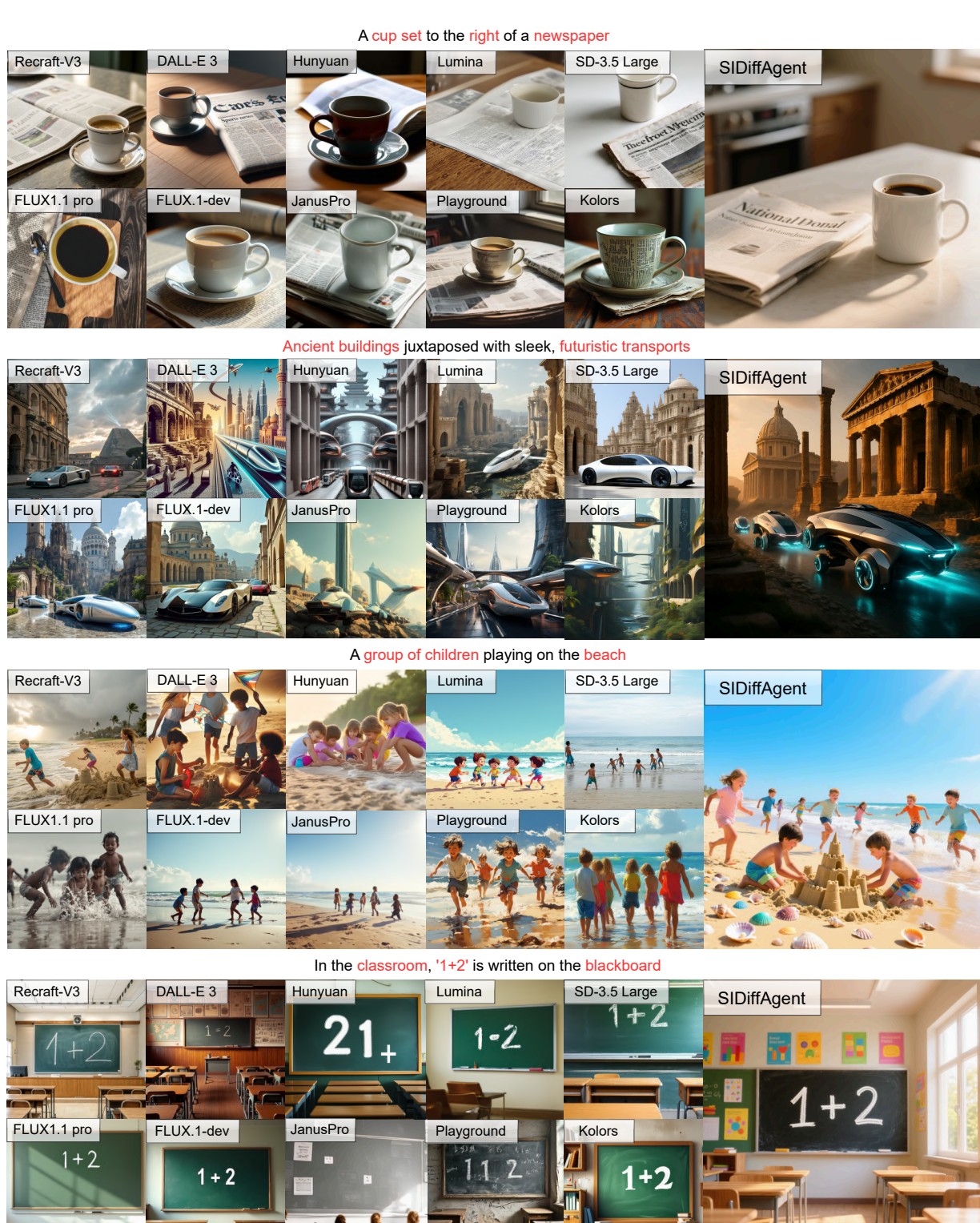

Figure 7: Qualitative comparison between multiple state-of-the-art method and *SIDiffAgent* on GenAI-Bench

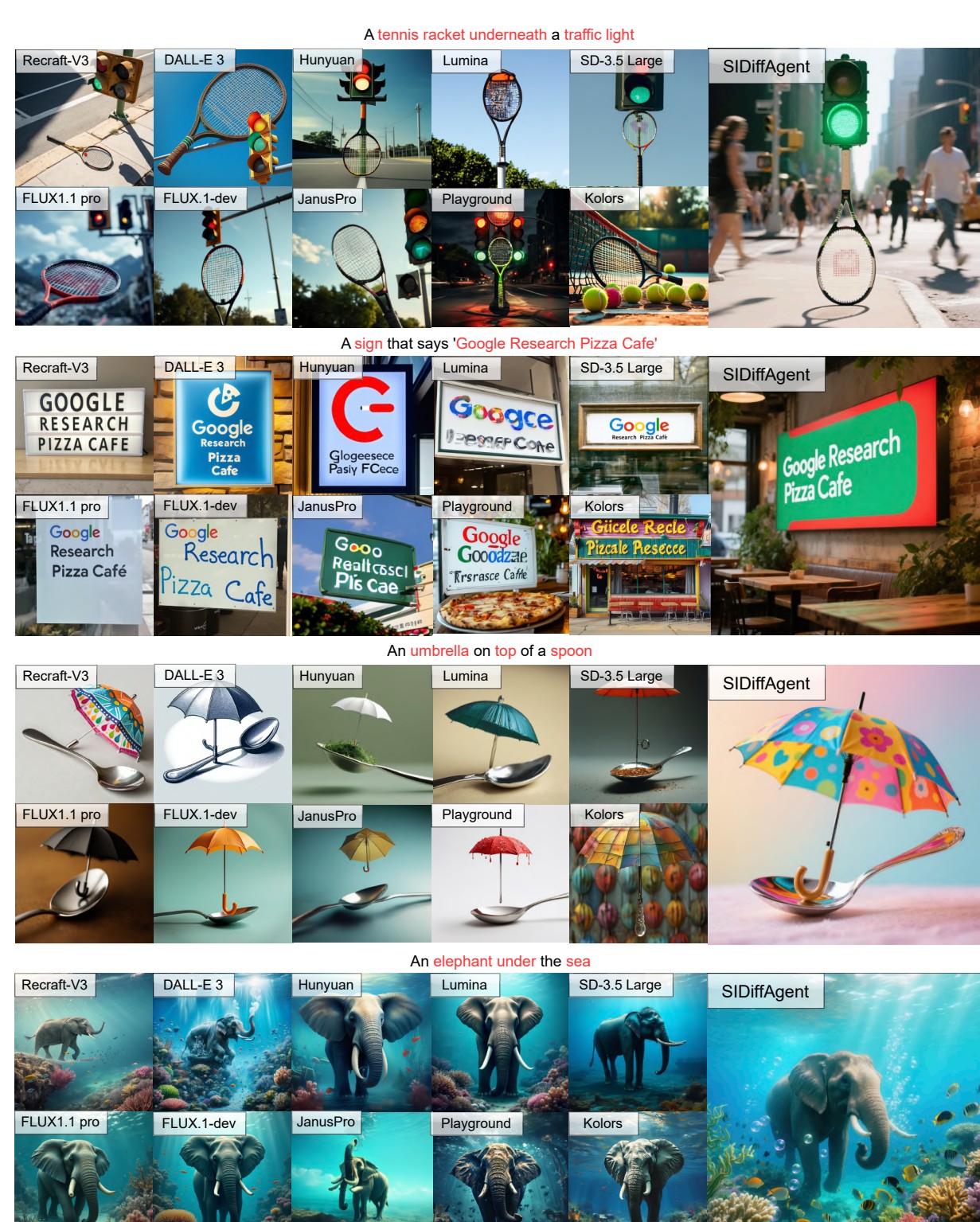

Figure 8: Qualitative comparison between multiple state-of-the-art method and *SIDiffAgent* on DrawBench

**Examples of images from SIDiffAgent Episode 2 vs SIDiffAgent (Drawbench)**

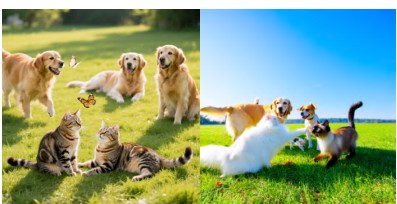
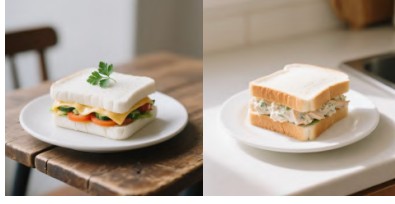
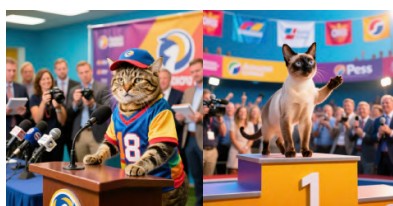

Two cats and three dogs sitting
on the grass.

A white colored sandwich.

Photo of an athlete cat explaining it's latest
scandal at a press conference to journalists.

**Examples of images from SIDiffAgents vs Qwen-Agents (Drawbench)**

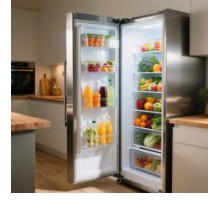
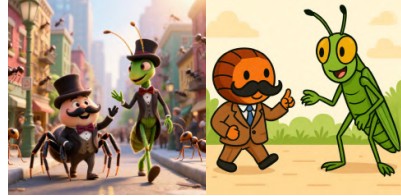
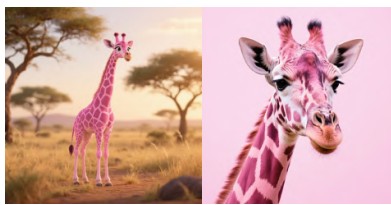

An appliance or compartment which is
artificially kept cool and used to store food
and drink.

A spider with a moustache bidding an
equally gentlemanly grasshopper a good
day during his walk to work.

A pink colored giraffe.

**Examples of images from SIDiffAgent Episode 2 vs SIDiffAgent (GenAIBenchmark)**

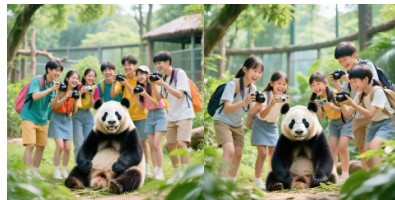
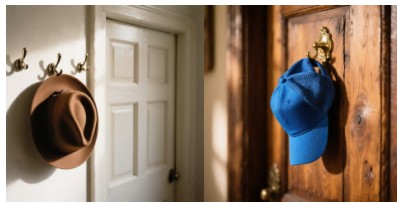
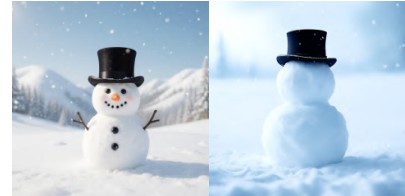

A group of students gathered around a panda,
all with digital cameras in their hands.

A hat hanging on a hook to the left of a door.

A snowman with a hat, and no scarf
around its neck.

**Examples of images from SIDiffAgents vs Qwen-Agents (GenAIBenchmark)**

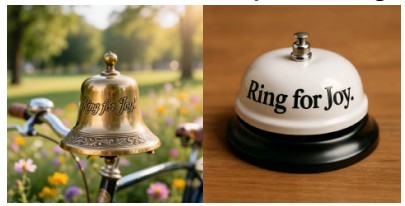
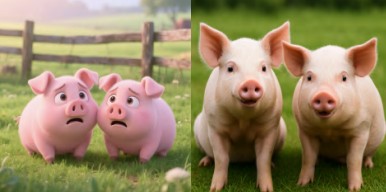
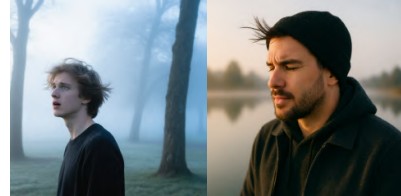

A bicycle bell engraved with 'Ring for Joy.'

Two anxious pigs.

A man without a beanie, feeling the chilly
morning air directly on his hair.

Figure 9: Qualitative comparison between various ablations described in Section 4.2

