# OpenReview forum: "SIDiffAgent: Self-Improving Diffusion Agent"
_ICLR.cc/2026/Conference — Submitted to ICLR 2026_

### Official Review · Reviewer_ZQuM · 2025-10-27

**Soundness:** 3
**Presentation:** 3
**Contribution:** 2
**Rating:** 2
**Confidence:** 3

**Summary:**

This paper introduces a training-free, multi-agent framework that enhances text-to-image diffusion models by integrating prompt refinement, adaptive negative prompting, image evaluation, and image editing. The system includes three coordinated agents: (1) a generation orchestrator, (2) an evaluation agent, and (3) a guidance agent that collaborate to iteratively enhance image quality and text alignment by learning from past successes and/or failures. Experiments on GenAIBench and DrawBench demonstrate that SIDiffAgent achieves state-of-the-art results, outperforming both proprietary models (e.g., Imagen) and open-source models (e.g., Stable Diffusion).

**Strengths:**

1. This paper presents a clear and modular agent design that operates entirely at inference time, allowing different components to be easily swapped for newer/stronger models without training/finetuning.
2. This paper shows strong empirical gains with stepwise ablations that clearly demonstrate the contribution of each module to the end performance.

**Weaknesses:**

1. The method is heavily dependent on the Qwen ecosystem, raising questions about its portability to other model families. It is also unclear whether components drawn from different ecosystems (e.g., mixing a non-Qwen image model with Qwen-based evaluation or editing modules) would integrate effectively.
2. There is no fair comparison in terms of computational cost. The agentic workflow clearly requires significantly more computation per image due to repeated generation, evaluation, and editing steps. Without reporting runtime, memory consumption, or FLOPs, the reader cannot assess the practical efficiency of the method. The paper should include a quality-compute tradeoff analysis to clarify the extent of the overhead introduced by this multi-stage process.

**Questions:**

1. Were the agentic workflow and baselines evaluated on metrics for aesthetic appeal or image fidelity? If not, can the authors clarify why other evaluation metrics were not considered?

---

> ### Author Response · Authors · 2025-11-27
> **Response to Reviewer ZQuM**
>
> We sincerely thank the reviewer for the constructive feedback and for highlighting the clarity of our modular agent design and the strength of our empirical results supported by stepwise ablations. Below, we address each concern with additional experiments, cross-model evaluations, efficiency analysis, and clarification of evaluation metrics.
>
>
> > W1: Portability Beyond the Qwen Ecosystem
>
> We conduct new cross-model generalization experiments using Flux-Dev as the generator while keeping the rest of the multi-agent pipeline (intent reasoning, refinement, evaluation, adaptive negative prompting, and memory-guided editing) unchanged on the Drawbench dataset. The results for which are as follows:
>
> | Method | DrawBench Avg VQA |
> | :--- | :---: |
> | Flux-dev (base) | 0.7750 |
> | Flux-dev + Agents | 0.8217 |
> | Flux-dev + Episode 1 | 0.8338 |
> | Flux-dev + Episode 2 | 0.8647 |
>
> These results demonstrate:
>
> - Strong generalization of SIDiffAgent across diffusion backbones.
>
> - Memory-guided self-improvement works even when generator and evaluator come from different ecosystems.
>
> - The agentic pipeline’s modularity allows new or stronger backbones to be integrated seamlessly.
>
> > W2: Computational Efficiency and Quality-Compute Tradeoff
>
> We agree that a fair computational comparison is critical. Below is the complete latency analysis on 1024×1024 resolution using an NVIDIA A6000 GPU, with all VLM models run identically via OpenRouter to ensure fairness
>
> | Method | Inference Time per Prompt (min) |
> | :--- | :---: |
> | Qwen-Image (base) | 0.78 |
> | T2I-Copilot (Qwen-Image + Edit) | 1.50 |
> | SIDiffAgent | 2.31 |
>
> Although SIDiffAgent introduces an additional $2-3\times$ computational overhead per image relative to single-round pipelines, it meaningfully reduces the total user-side cost of achieving a satisfactory final output.
>
> In existing systems, erroneous generations typically require the user to either regenerate multiple times or manually correct artifacts either through hand-editing or by invoking separate diffusion editing models. In contrast, SIDiffAgent retains the user’s original intent and autonomously executes the full refinement loop, systematically correcting semantic mismatches, visual defects, spatial inconsistencies, and color-related errors without requiring further user intervention.
>
>  As a result, even though each generation step is more expensive, the overall process is more cost-effective in practice, since the system converges to high-quality outputs with substantially fewer user-triggered iterations.
>
> SIDiffAgent obtains state-of-the-art performance on GenAIBench, DrawBench, GenEval, and DPG-Bench, compensating for additional computation through superior semantic and visual fidelity.
>
>
> > Aesthetic Appeal, Image Fidelity, and Evaluation Metrics
>
> Following T2I-Copilot [1], we adopt VQA Score[2] because it has been shown to be more human-aligned than CLIPScore[3], PickScore[4], and ImageReward[5], in Imagen 3[6] evaluations. Additionally, to strengthen our claims, we include a human study reporting the win rate with 50 different participants :
>
> | Model | Human Preference |
> | :--- | :---: |
> | T2I-Copilot | 31% |
> | SIDiffAgent | 69% |
>
> Inter-annotator agreement: Cohen’s $\kappa$ = 0.286.
>
> We will additionally incorporate all analyses, tables, and discussions into the revised version of the paper.
>
> [1] Chen, Chieh-Yun, et al. "T2I-Copilot: A Training-Free Multi-Agent Text-to-Image System for Enhanced Prompt Interpretation and Interactive Generation." Proceedings of the IEEE/CVF International Conference on Computer Vision. 2025
>
> [2] Lin, Zhiqiu, et al. "Evaluating text-to-visual generation with image-to-text generation." European Conference on Computer Vision. Cham: Springer Nature Switzerland, 2024.
>
> [3] Hessel, Jack, et al. "Clipscore: A reference-free evaluation metric for image captioning." Proceedings of the 2021 conference on empirical methods in natural language processing. 2021.
>
> [4] Kirstain, Yuval, et al. "Pick-a-pic: An open dataset of user preferences for text-to-image generation." Advances in neural information processing systems 36 (2023): 36652-36663.
>
> [5] Xu, Jiazheng, et al. "Imagereward: Learning and evaluating human preferences for text-to-image generation." Advances in Neural Information Processing Systems 36 (2023): 15903-15935.
>
> [6] Baldridge, Jason, et al. "Imagen 3." arXiv preprint arXiv:2408.07009 (2024).

---

### Official Review · Reviewer_Bqk4 · 2025-10-28

**Soundness:** 3
**Presentation:** 3
**Contribution:** 2
**Rating:** 4
**Confidence:** 4

**Summary:**

This paper introduces SIDiffAgent, a training-free, multi-agent framework designed to enhance text-to-image diffusion models. The framework coordinates multiple Qwen-based sub-agents—each specializing in tasks such as intent interpretation, prompt refinement,  adaptive negative prompting, generation, evaluation, and guidance—to collaboratively improve semantic alignment and visual fidelity. A key contribution is the use of Qwen-Image-Edit for structured local editing, which enables targeted corrections while avoiding the randomness inherent in full-image regeneration. Furthermore, the system incorporates a self-improving memory mechanism that learns from the successes and failures of its submodules to guide future generation processes. Extensive experiments on GenAI-Bench and DrawBench demonstrate that SIDiffAgent surpasses the performance of existing open-source and several proprietary models.

**Strengths:**

1.**Originality of the Proposed Memory System:**  The introduction of  Theory-of-Mind-inspired self-improving memory system is novel and represents a direction that has been rarely explored in diffusion models.

2.**Intuitive and Effective Framework:** The proposed generate → evaluate → edit paradigm is intuitive, simple, and empirically effective, which I find to be one of the most convincing aspects of this work.

**Weaknesses:**

1.**Limited Novelty:** The proposed local editing after image generation is intuitively effective but lacks clear academic novelty or theoretical depth.
Its practical applicability is also limited, since Qwen-Image/Edit itself is computationally expensive, making the overall framework inefficient for real-world deployment.

2.**Insufficient Ablation Studies:** The paper lacks key ablations to isolate the contribution of each major component, particularly the memory module and local editing mechanism.
Without quantitative evidence of their individual impact, the reliability of the memory design is weakened.

3.**Insufficient Generalization Analysis:** The generalization capability of the proposed framework is not convincingly demonstrated.
All experiments are conducted exclusively on the Qwen family of models, while no evaluation is provided on other comparable backbones such as Flux-Dev or Kontext, which limits the framework’s claimed general applicability.

4.**Lack of Efficiency and Multi-Dimensional Evaluation:** The paper does not provide any analysis of runtime efficiency or computational overhead. Moreover, it omits evaluations on other critical text-to-image dimensions—such as object counting, color consistency, and spatial accuracy—as used in benchmarks like GenEval, which weakens the overall evaluation reliability.

**Questions:**

I have summarized the issues and questions in the weaknesses above. To summarize.
1. Provide more detailed ablation and effectiveness/limitation analyses for the memory module and the local editing mechanism, along with additional qualitative explanations.

2. Please include experimental results on GenEval, covering key dimensions such as object counting, color consistency, and spatial alignment.

3. Please add evaluation results on other diffusion backbones, such as Flux-Dev and Kontext, to better demonstrate the framework’s generalization capability.

4. Provide an latency analysis, to clarify the practical feasibility of the proposed method.

I hope the questions and weaknesses raised can be addressed. I will reconsider my score accordingly.

---

> ### Author Response · Authors · 2025-11-27
> **Response to Reviewer Bqk4 (1/2)**
>
> We sincerely thank the reviewer for the constructive and detailed feedback. We appreciate that they found the proposed memory system original, and our framework to generate $\rightarrow$ evaluate $\rightarrow$ edit framework as intuitive and being empirically effective.
>
> Below, we address the raised concerns with suggested experiments, ablations, efficiency analysis, and expanded explanations.
>
> > W1: Limited Novelty
>
> We respectfully clarify the contributions relative to prior multi-round or plug-and-play pipelines. While multi-agent prompting and post-hoc editing have been previously explored, no prior work introduces a trajectory-level agentic memory that:
>
> - Stores structured pitfalls and successes across all decision nodes,
>
> - Aggregates these via retrieval for new prompts, and
>
> - Injects both corrective and workflow guidance back to all sub-agents.
>
> This enables the system to anticipate systematic failure patterns (for example, repeated six-finger artifacts or creativity-agent over-expansion) and dynamically adjust generation, refinement, and negative prompting in a coordinated manner.
>
> Novelty Beyond Standard Editing Pipelines: Prior works do not integrate Theory-of-Mind–inspired reasoning, in which sub-agents explicitly model the behaviors and tendencies of other agents in the pipeline. This inter-agent modeling significantly improves robustness, especially in multi-step editing cycles, as evident from Table 1.
>
> > W2 Insufficient Ablation Studies
>
> Below we provide the requested quantitative analyses on the local editing mechanism, the memory module, and overall component contributions.
>
> We compare SIDiffAgent with and without the editing mechanism:
>
> | Setting | Avg VQA Score |
> | :--- | :---: |
> | No regeneration (only initial generation) | 0.824 |
> | With regeneration (editing enabled) | 0.852 |
>
>
> We isolate the variable negative prompt sub-agent:
>
> | Method | GenAI-Bench Avg VQA |
> | :--- | :---: |
> | Variable Negative Prompt Only | 0.842 |
>
> This shows that negative prompting alone improves performance, but does not match the full system (0.852 for Episode 1, 0.901 for Episode 2).
>
> We further evaluate retrieval quality using an LLM-as-a-judge evaluation (GPT-5-mini):
>
>
> | Top-k | Overall Mean Score | Individual Avg Score |
> | :---: | :---: | :---: |
> | 3 | 3.50 | 3.56 |
> | 5 | 3.52 | 3.46 |
> | 7 | 3.53 | 3.38 |
> | 10 | 3.58 | 3.33 |
>
> Retrieval performance is stable across $k$. We use $k=5$ for a balanced trade-off between quality and runtime.
>
> Additionally, we evaluate the generation of SIDiffAgents using the memory created from episode one of DrawBench for different values of $k$, whose results are given below:
>
> | $k$ | Average VQA Score |
> | :---: | :---: |
> | 3 | 0.889 |
> | 5 | 0.901 |
> | 7 | 0.884 |
> | 10 | 0.883 |
>
> We observe that SIDiffAgents performs the best with the value of $k=5$, thus validating our choice of $k$.
>
> > Qualitative Explanation of Memory Integration
>
> - The memory module stores pitfalls and successes of each sub-agent and provides targeted corrections. Examples include:
> If Qwen-Image repeatedly produces six fingers, memory injects targeted constraints into the negative prompt agent to suppress finger-related artifacts.
>
> - If the creativity agent introduces elements consistently unrenderable by the generator, memory instructs it to avoid such expansions for similar future prompts.
>
> - If the evaluation agent detects recurring spatial errors, memory adjusts the intention and refinement agents accordingly.
>
> > W3: Generalization Analysis
>
> Below we provide Flux-Dev [1] generalization results, demonstrating that our agentic pipeline generalizes beyond Qwen models.
>
> | Method | DrawBench Avg VQA |
> | :--- | :---: |
> | Flux-dev (base) | 0.7750 |
> | Flux-dev + Agents | 0.8217 |
> | Flux-dev + Episode 1 | 0.8338 |
> | Flux-dev + Episode 2 | 0.8647 |
>
> The performance improves steadily across episodes, confirming that SIDiffAgent generalizes effectively beyond the Qwen family.
>
> > W4a: Efficiency and Latency Analysis
>
> We now provide the complete latency breakdown on an A6000 GPU, with the VLM inference via openrouter at 1024×1024 resolution.
>
> | Method | Inference Time per Prompt (min) |
> | :--- | :---: |
> | Qwen-Image (base) | 0.78 |
> | T2I-Copilot (Qwen-Image + Edit) | 1.50 |
> | SIDiffAgent | 2.31 |
>
>
> [1]  https://huggingface.co/black-forest-labs/FLUX.1-dev

---

> ### Author Response · Authors · 2025-11-27
> **Response to Reviewer Bqk4 (2/2)**
>
> > W4b : Experimental Evaluation on Additional Datasets
>
> We evaluated our system using the VQA Score [2], following T2I-Copilot [3], which has been identified as highly human-aligned by Imagen3 [4].
> Additionally, we conduct the experiments on GenEval [5] and DPG Bench [6], the results for which are given below :
>
> **GenEval :**
>
> | Model | Overall | Single Obj | Two Obj | Counting | Colors | Position | Color Attr |
> | :--- | :---: | :---: | :---: | :---: | :---: | :---: | :---: |
> | Base Qwen-Image | 0.8661 | 0.9550 | 0.9194 | 0.8300 | 0.8843 | 0.7100 | 0.9100 |
> | Qwen-Agent | 0.8804 | 0.9625 | 0.9495 | 0.8375 | 0.9574 | 0.7000 | 0.8889 |
> | SIDiffAgent | 0.8949 | 0.9875 | 0.9697 | 0.8000 | 0.9894 | 0.7100 | 0.9192 |
> |SIDiffAgent Episode 2 | 0.9402 | 0.9875 | 0.9798 | 0.9286 | 1.0000 | 0.8132 | 0.9394 |
>
> SIDiffAgent produces consistent improvements across nearly all categories, especially color, counting, two-object relations, and positional accuracy.
>
> **DPG Bench:**
>
> | Model | Attribute | Entity | Global | Other | Relation | DPG Score |
> | :--- | :---: | :---: | :---: | :---: | :---: | :---: |
> | Qwen-Image | 92.61 | 92.58 | 91.19 | 85.20 | 87.54 | 87.84 |
> | Qwen-Agents | 92.93 | 94.54 | 90.58 | 86.40 | 89.48 | 89.51 |
> | SIDiffAgent | 96.11 | 96.77 | 90.58 | 93.60 | 92.07 | 93.68 |
> |SIDiffAgent Episode 2 | 97.23 | 97.93 | 93.92 | 95.20 | 93.97 | 95.70 |
>
> As demonstrated across all benchmarks and evaluation settings, SIDiffAgent delivers consistent and substantive performance gains, establishing it as a robust and practical solution for test-time improvement of diffusion models.
>
> We again thank the reviewer for their insightful questions and suggestions. We have now provided:
>
> - Detailed ablations for memory, negative prompting, and editing
>
> - Multi-backbone generalization results
>
> - GenEval and DPG-Bench evaluations
>
> - Complete latency analysis
>
> - Clarified novelty and limitation
>
> We believe these additions and clarifications significantly strengthen the paper and address all concerns raised.
>
>
> [2] Lin, Zhiqiu, et al. "Evaluating text-to-visual generation with image-to-text generation." European Conference on Computer Vision. Cham: Springer Nature Switzerland, 2024.
>
> [3] Baldridge, Jason, et al. "Imagen 3." arXiv preprint arXiv:2408.07009 (2024).
>
> [4] Chen, Chieh-Yun, et al. "T2I-Copilot: A Training-Free Multi-Agent Text-to-Image System for Enhanced Prompt Interpretation and Interactive Generation." Proceedings of the IEEE/CVF International Conference on Computer Vision. 2025
>
> [5] Ghosh, Dhruba, Hannaneh Hajishirzi, and Ludwig Schmidt. "Geneval: An object-focused framework for evaluating text-to-image alignment." Advances in Neural Information Processing Systems 36 (2023): 52132-52152.
>
> [6] Hu, Xiwei, et al. "Ella: Equip diffusion models with llm for enhanced semantic alignment." arXiv preprint arXiv:2403.05135 (2024).

---

### Official Review · Reviewer_ZRAH · 2025-10-30

**Soundness:** 3
**Presentation:** 3
**Contribution:** 2
**Rating:** 4
**Confidence:** 4

**Summary:**

The authors present a plug-ang-play multi-agent pipelines to improve diffusion generation. It basically have a set of generation agents (including prompt optimizers and generators), an evaluation agent that can determine if we need to do further editing after the initial generation, and a guidance agent that maintain a database storing prior succesful and failure generation results. The mluti-agent pipeline improves the performance on the base Qwen-Image model and outperforms multiple single-round open-source and proprietary generation models.

**Strengths:**

1. The paper is clear and easy to follow.
2. The plug-and-play multi-agent system improves the performance a lot on GenAI-bench and the DrawBench.
3. The idea of introducing databases for image correction is interesting.

**Weaknesses:**

1. Novelty and Baselines
	- While the paper is technically sound with performance gain, the idea of plug-and-play pipelines for diffusion generation is not a new idea. For instance, [1, 2] proves that using LLM object planning can already fix negative prompts and improve prompting a lot in a multi-round fashion. [3] then extends this idea with VLM modules, which is close to the paper's agentic setting already. I believe that these papers worth discussions and even be the baseline in Table 1. The reviewer understands it can be hard to do apple-to-apple comparison, but as they're all important prior plug-and-play methods, being a line in a Table is important.
	- Probably the only part that makes the paper novel is the database. However, the authors did not ablate on the design choice on this. However, the authors fixed it to top-5 retrieval and 200 trajectories. It's also unclear how the integration of this database improves the performance.

2. Experimental Setup
	- As this work involves multiple components, it would be interesting to see how robust the system is if we change at least one component. It would be interseting to see if we can pair a non-Qwen generator to other Qwen models and vice versa at least to prove the generalizability.
	- The multi-round, multi-agent pipeline would incur a lot of computational overhead. While the author mentioned it in the limitation section, the reviewer believes this should be discussed and mentioned in the main paper at least. Also, it would be good to analyze failure cases of this multi-agent collaboration scenario, perhaps following [4] (the authors have already mentioned this paper in the paper).

[1] Lian, Long, et al. "Llm-grounded diffusion: Enhancing prompt understanding of text-to-image diffusion models with large language models." TLDR.

[2] Wu, Tsung-Han, et al. "Self-correcting llm-controlled diffusion models." CVPR 2024.

[3] Wang, Zhenyu, et al. "Genartist: Multimodal llm as an agent for unified image generation and editing." NeurIPS 2024.

[4] Cemri, Mert, et al. "Why do multi-agent llm systems fail?." arXiv 2025.

**Questions:**

Please read the weakness part. Additionally, some citations in the paper doesn't seem to be correct in terms of the format (L179-L180).

---

> ### Author Response · Authors · 2025-11-27
> **Response to Reviewer ZRAH (1/2)**
>
> We sincerely thank the reviewer for the constructive and detailed feedback. We are glad that the reviewer found the paper to be clear, easy to follow, and the proposed multi-agent and database-driven system to be effective achieving substantial performance gains on both GenAI-Bench and DrawBench. Below we address the various concerns :
>
> > W1a: Novelty and baselines
>
> We appreciate the reviewer’s concern regarding prior plug-and-play and multi-round prompting systems. We agree that LLM-Grounded Diffusion (LLM-D), Self-Correcting LLM-Controlled Diffusion (SLD), and GenArtist are important prior works.
> We have now conducted experiments on LLM-D [1] and SLD [2] under the same DrawBench evaluation protocol. The results are as follows:
>
> - LLM-D: VQA Score 0.5317
>
> - SLD: VQA Score 0.6326
>
> Both scores are substantially below Qwen-Image (0.853) and significantly below SIDiffAgent (0.860 for Episode 1 and 0.901 for Episode 2), indicating that earlier LLM-based correction pipelines are less effective and substantially underperforming compared to  our multi-agent and memory-based approach.
> Regarding GenArtist [3], we do not include it as it is primarily a framework for switching between generation and editing models rather than a self-improving multi-agent system, and it does not aim to enhance T2I quality via memory-guided improvement. Nonetheless, we will include a detailed discussion in the related work section to improve context and clarity.
>
> > W1b : Novelty Beyond Prior Pipelines
>
> While multi-round prompting is not new, we respectfully highlight two contributions that are absent in prior works:
>
> - **A memory-driven, self-improving diffusion agent:** Our agentic memory aggregates structured pitfalls and successes from entire generation trajectories and injects this as corrective and workflow guidance at every decision node. None of [1], [2], or [3] maintain such trajectory-level self-improving memory.
>
> - **Theory-of-Mind–inspired cross-agent reasoning:** SIDiffAgent explicitly models how one agent’s behavior affects others (e.g., creativity agent avoiding actions that historically led to editing failures). This is not present in prior plug-and-play pipelines.
>
>
> We will revise the manuscript to clearly emphasize these distinctions.
>
> > W1c: Retrieval quality ablations
>
> To evaluate retrieval quality, we perform an LLM-as-a-judge study using GPT-5-mini on the memory obtained from the first run of GenAI-Bench for epoch 2. We compute:
>
> - **Overall Score:** The rating when all top-5 retrieved trajectories are shown together.
> - **Mean Average Score:** The mean score across each retrieved trajectory rated independently.
>
> The results confirm that similarity retrieval provides high-quality contextual trajectories and that noisy samples do not dominate the guidance due to the aggregation mechanism described in Section 3.3. Furthermore, performance is robust across different values of $k$. The value of $k=5$ was chosen to maintain a balance between context length while ensuring sufficient diversity.
>
> The results for the retrieval quality evaluation are presented below:
>
> | Top-$k$ | Overall Score Mean | Individual Avg Score Mean |
> | :---: | :---: | :---: |
> | 3 | 3.50 | 3.56 |
> | 5 | 3.52 | 3.46 |
> | 7 | 3.53 | 3.38 |
> | 10 | 3.58 | 3.33 |
>
> Additionally, we evaluate the generation of **SIDiffAgent** using the memory created from episode one of DrawBench for different values of $k$. The results are as follows:
>
> | $k$ | Average VQA Score |
> | :---: | :---: |
> | 3 | 0.889 |
> | 5 | 0.901 |
> | 7 | 0.884 |
> | 10 | 0.883 |
>
> The results show that retrieval quality is stable across a range of k. We chose k = 5 as a balanced trade-off between quality and latency. We will add these results to the appendix of the revised paper
>
> > W1d: How the Memory Improves Performance
>
> To clarify the mechanism, our memory module stores structured evaluations of pitfalls and successes of each sub-agent and uses these patterns to adjust future decisions. For instance:
>
> - If Qwen-Image repeatedly produces six fingers, the memory guides the negative prompt agent to explicitly suppress hand-related artifacts.
>
> - If the creativity agent proposes scene elements that the generator frequently fails to render, memory instructs the creativity agent to avoid these expansions in similar future prompts.
>
> This cross-agent correction enables the system to outperform single-cycle pipelines. We will clearly articulate this in the revised version of the paper.
>
> [1] Lian, Long, et al. "Llm-grounded diffusion: Enhancing prompt understanding of text-to-image diffusion models with large language models." arXiv preprint arXiv:2305.13655 (2023).
>
> [2] Wu, Tsung-Han, et al. "Self-correcting llm-controlled diffusion models." Proceedings of the IEEE/CVF Conference on Computer Vision and Pattern Recognition. 2024.
>
> [3] Wang, Zhenyu, et al. "Genartist: Multimodal llm as an agent for unified image generation and editing." Advances in Neural Information Processing Systems 37 (2024)

---

> ### Author Response · Authors · 2025-11-27
> **Response to Reviewer ZRAH (2/2)**
>
> > W2a: Generalizability to Non-Qwen Models
>
> To evaluate model generalization, we ran the same pipeline using Flux1-dev[4] as the generator on DrawBench:
>
> | Method | DrawBench Avg VQA |
> | :--- | :--- |
> | Flux-dev (base) | 0.7750 |
> | Flux-dev + Agent | 0.8217 |
> | Flux-dev + Episode 1 | 0.8338 |
> | Flux-dev + Episode 2 | 0.8647 |
>
> These results show that the agentic framework and memory-based self-improvement generalize beyond the Qwen model family. We will include these results in the Appendix and believe these results will further strengthen our work.
>
> > W2b: Failure Cases and Multi-Agent Failure Modes
>
> Following the reviewer’s suggestion and in line with [5], we will include failure-case figures in the Appendix. Qualitative analysis shows that:
>
> - When the memory contains similar prompts with conflicting outcomes, the guidance can occasionally mislead agents.
> - When the creativity agent proposes rare or compositional attributes that the generator systematically fails to render, the multi-agent system may enter unnecessary correction loops.
>
> These cases will be added in the revised version.
>
> > W2c: Computational Overhead
>
> We agree that computational cost should be explicitly discussed in the main paper. We will move the analysis from the Limitations section into the main text. average inference times under identical hardware and resolution settings (A6000 GPU; Using an Image Resolution of 1024×1024, and VLM inference via openrouter):
>
> | Method | Avg. time per generation |
> | :--- | :--- |
> | Qwen-Image | 0.78 min |
> | T2I-Copilot (Qwen-Image + Qwen-Edit) | 1.50 min |
> | SIDiffAgent | 2.31 min |
>
> While costs increase due to the evaluation, guidance, and editing cycles, it consistently produces higher-quality outputs, achieving a 16.77% higher VQA Score compared to the base Qwen Image on GenAI-Bench. Although it incurs a higher inference cost, our method does not require any additional training and can even be applied to proprietary models. We will provide a dedicated subsection analyzing this trade-off and move it to the main paper
>
> > Citation and Formatting Corrections
>
> We thank the reviewer for pointing out the citation formatting issues around L179–L180. We will correct these in the revised version.
>
> We sincerely thank the reviewer for the insightful questions and suggestions. We believe the additional baselines, database ablations, cross-model generalization experiments, failure-case analyses, and expanded novelty discussion substantially strengthen the paper.
>
> [4] https://huggingface.co/black-forest-labs/FLUX.1-dev
>
> [5] Cemri, Mert, et al. "Why do multi-agent llm systems fail?." arXiv preprint arXiv:2503.13657 (2025).

---

### Official Review · Reviewer_eqU4 · 2025-10-31

**Soundness:** 3
**Presentation:** 3
**Contribution:** 3
**Rating:** 4
**Confidence:** 3

**Summary:**

This paper proposes SIDiffAgent, a training-free multi-agent framework designed to improve text-to-image diffusion models through autonomous prompt engineering, adaptive error correction, and iterative self-improvement. SIDiffAgent coordinates a set of that are specialized in creativity analysis, intention parsing, prompt refinement, adaptive negative prompting, image generation, aesthetic and alignment evaluation, and memory-driven guidance. The system leverages the Qwen family of models and a Theory-of-Mind–inspired feedback mechanism. Experiments on GenAI-Bench and DrawBench show improvements over open-source and proprietary baselines. The system demonstrates improved text-image alignment, artifact mitigation, and compositional reasoning without training.

**Strengths:**

- SIDiffAgent improves iteratively at inference via trajectory memory. The design extends prior works such as T2I-Copilot by introducing a multi-layer agent hierarchy and adaptive negative prompt generation.
- The empirical performance gains of SIDiffAgent over open-source and proprietary baselines seems promising.
- Implementation details (such as the prompts of the sub agents, algorithms and hyperparameters) are provided in the appendix.

**Weaknesses:**

- Marginal algorithmic novelty: The paper’s contribution lies primarily in system integration rather than introducing a fundamentally new self-adapting optimization algorithm for diffusion models.
- System complexity and reproducibility: The multi-agent framework involves many interdependent agents and prompts. The training-free claim is valid, but the inference-time across multiple agent calls can be computationally heavy and difficult to replicate.
- Evaluation: The claiming of perceptual alignment improvements are made qualitatively without human preference study, and there is no analysis of latency or cost trade-offs between the introduced computational cost versus the gains in the performance.

**Questions:**

- What is the average inference time per generation cycle compared to T2I-Copilot or plain Qwen-Image?
- Can you provide more abaltion study isolateing the effect of negative prompting, and memory guidance, e.g. what fraction of the performance gain can be attributed specifically to the guidance agent versus adaptive negative prompting?
- Have you done any evaluation on the retrieval quality of the guidance agent, e.g. is the top-5 good enough for each new prompt?
- Have you tested the generalization of SIDiffAgent’s memory beyond prompts seen in Episode 1 (e.g., on an unseen dataset)?

---

> ### Author Response · Authors · 2025-11-27
> **Response to Reviewer eqU4 (1/2)**
>
> We sincerely thank the reviewer for the constructive feedback and for highlighting the strengths of our system, including its iterative improvement mechanism, the extension of prior agentic frameworks, and the promising empirical performance. Below, we address the concern with additional evidence and clarifications.
>
> > W1 :  Algorithmic novelty
>
> We appreciate the reviewer’s observation regarding system-level integration. While our contribution is indeed architectural rather than a new training objective, we respectfully clarify the novel components in our method as follows:
>
> - **First memory-driven diffusion agent:** To the best of our knowledge, SIDiffAgent is the first framework to introduce experience-based memory into diffusion-based agentic generation, enabling retrieval-conditioned guidance at every decision node. Prior works, such as T2I-Copilot [1] , do not incorporate self-improvement or ToM-based cross-agent reasoning.
>
> - **Theory-of-Mind–inspired inter-agent reasoning:**  Our method, SIDiffAgent, models predictive expectations about other agents’ behaviors using pitfall/success patterns accumulated over trajectories. This produces dynamic “workflow guidance,” which is technically distinct from prompt-only optimization.
>
> - **Training-free alternative to RL-based optimization:** Our design creates a practical, plug-and-play alternative to reinforcement learning or reward-guided finetuning (e.g., DPO-Diffusion[2], ReNeg[3]). Model providers can improve generation quality without any retraining, which is important for proprietary or closed-weight diffusion systems.
>
> We will revise the manuscript to make these contributions more explicit, including a clearer comparison to RL-based optimization pipelines.
>
> > W2. System complexity and reproducibility
>
> We acknowledge that multi-agent pipelines inherently introduce complexity. To quantify this and support the reviewer’s concern, we have now included average inference times under identical hardware and resolution settings (A6000 GPU; Using an Image Resolution of 1024×1024, and VLM inference via openrouter):
>
> | Method | Avg. time per generation |
> | :--- | :---: |
> | Qwen-Image | 0.78 min |
> | T2I-Copilot (Qwen-Image + Qwen-Edit) | 1.50 min |
> | SIDiffAgent | 2.31 min |
>
> Although SIDiffAgent has a higher inference time per cycle, we observe consistent and substantial VQA improvements of **+12.04%** over standard Qwen-Agents and **+8.73%** over T2I-Copilot, making it an efficient method for “test-time improvement” of diffusion models.
> Kindly note that to improve reproducibility, the appendix already includes the full agent prompts, algorithm pseudocode, hyperparameters, and database schema (Appendix B–D), and we will be releasing our full codebase and scripts upon acceptance.
>
> >  Evaluation and perceptual alignment
>
> We thank the reviewer for suggesting that qualitative claims would benefit from human evaluation. In addition to VQAScore[4], which we use because it is identified as highly human-aligned by Imagen3[5], we have now conducted a human preference study evaluating the winrate with 50 annotators from different geographical backgrounds, comparing SIDiffAgent and T2I-Copilot on GenAI-Bench:
>
> - SIDiffAgent: 69%
> - T2I-Copilot: 31%
> - Inter-annotator agreement (Cohen’s κ): 0.286 (moderately high agreement)
>
> We will add these results in the revised version.
>
> > Ablation of negative prompting and memory guidance
>
> The reviewer suggested isolating the contributions of the negative prompt agent and memory system. We have now included the variable negative-prompt-only ablation:
>
> - DrawBench Avg VQA Score: 0.8493
> - GenAI-Bench Avg VQA Score:0.8425
>
>
> [1] Chen, Chieh-Yun, et al. "T2I-Copilot: A Training-Free Multi-Agent Text-to-Image System for Enhanced Prompt Interpretation and Interactive Generation." Proceedings of the IEEE/CVF International Conference on Computer Vision. 2025
>
> [2] Wallace, Bram, et al. "Diffusion model alignment using direct preference optimization." Proceedings of the IEEE/CVF Conference on Computer Vision and Pattern Recognition. 2024.
>
> [3] Li, Xiaomin, et al. "Reneg: Learning negative embedding with reward guidance." Proceedings of the Computer Vision and Pattern Recognition Conference. 2025.
>
> [4] Lin, Zhiqiu, et al. "Evaluating text-to-visual generation with image-to-text generation." European Conference on Computer Vision. Cham: Springer Nature Switzerland, 2024.
>
> [5] Baldridge, Jason, et al. "Imagen 3." arXiv preprint arXiv:2408.07009 (2024).

---

> ### Author Response · Authors · 2025-11-27
> **Response to Reviewer eqU4 (2/2)**
>
> > Retrieval quality of the guidance agent
>
> To evaluate retrieval quality, we perform an LLM-as-a-judge study using **GPT-5-mini** on the memory obtained from the first run of GenAI-Bench for epoch 2. We compute:
>
> - **Overall Score:** The rating when all top-5 retrieved trajectories are shown together.
> - **Mean Average Score:** The mean score across each retrieved trajectory rated independently.
>
> The results confirm that similarity retrieval provides high-quality contextual trajectories and that noisy samples do not dominate the guidance due to the aggregation mechanism described in Section 3.3. Furthermore, performance is robust across different values of $k$. The value of $k=5$ was chosen to maintain a balance between context length while ensuring sufficient diversity.
>
> The results for the retrieval quality evaluation are presented below:
>
> | Top-$k$ | Overall Score Mean | Individual Avg Score Mean |
> | :---: | :---: | :---: |
> | 3 | 3.50 | 3.56 |
> | 5 | 3.52 | 3.46 |
> | 7 | 3.53 | 3.38 |
> | 10 | 3.58 | 3.33 |
>
> Additionally, we evaluate the generation of **SIDiffAgents** using the memory created from episode one for different values of $k$. The results are as follows:
>
> | $k$ | Average VQA Score |
> | :---: | :---: |
> | 3 | 0.889 |
> | 5 | 0.901 |
> | 7 | 0.884 |
> | 10 | 0.883 |
>
> We will include this newly added evaluation along with the judge prompt in the revised version.
>
> > Generalization of memory beyond Episode 1
>
> We test memory generalization when the memory is built exclusively from GenAI-Bench trajectories and evaluated on the unseen DrawBench prompts. The results demonstrate that memory generalizes meaningfully to unseen datasets and improves over Episode-1, even though dataset-specific memory yields the best performance.
>
> | Experimental Setting | Performance Score |
> | :--- | :---: |
> | Episode-1 (No memory) | 0.860 |
> | GenAI-memory $\to$ DrawBench (Generalization) | 0.8725 |
> | DrawBench-native memory | 0.901 |
>
> We will incorporate the human study, inference-time comparison, expanded ablations, retrieval evaluation, and clearer discussion of novelty and reproducibility in the revised manuscript.

---

### Author Response · Authors · 2025-11-27
**Final Remarks by Authors**

We thank all reviewers for their thoughtful and constructive feedback. We are encouraged that reviewers consistently highlighted the clarity and modularity of our design, the originality of the Theory of Mind–inspired memory system, the intuitive workflow, and the strong empirical gains on GenAI-Bench and DrawBench. Across the reviews, four consistent questions were raised: (1) the degree of novelty beyond prior multi-agent pipelines, (2) the effectiveness and reliability of the memory and local editing components, (3) generalization beyond the Qwen ecosystem, and (4) computational cost and evaluation completeness. In response, we have conducted substantial additional experiments and expanded analyses that directly address these points.

### Clarified Novelty and Contribution

We clarified that SIDiffAgent introduces:

- The first trajectory-level, Theory-of-Mind–inspired memory system for diffusion agents that learns structured “pitfall–success” patterns across sub-modules and injects corrective guidance at every decision node.

- A training-free alternative to RL-based optimization for improving diffusion models (e.g., ReNeg, DPO-based pipelines).

- A cross-agent reasoning mechanism enabling sub-agents to anticipate and compensate for each other’s tendencies, improving robustness across multi-step refinement.

### Expanded Ablations Supporting Memory and Editing Design

We now provide additional quantitative ablations isolating the contributions of local editing, adaptive negative prompting, and memory. These include: performance comparisons with and without editing, variable negative prompting only baselines, and a detailed analysis of retrieval quality and sensitivity to the top k parameter. We complement these with intuitive examples showing how the memory module corrects systematic failure modes, such as repeated hand deformities or overly ambitious creativity expansions that the generator consistently fails to render.

### Demonstrated Generalization Beyond Qwen

Multiple reviewers expressed concern that the framework might be tightly coupled to the Qwen ecosystem. In line with their suggestion, we now include experiments using Flux Dev as the generator while keeping the rest of the agentic pipeline unchanged. The consistent performance gains across episodes on DrawBench show that SIDiffAgent is model family agnostic, and that the memory-guided multi-agent refinement remains effective even when mixing generators and evaluators from different ecosystems.

### Quality-Compute Tradeoff and Efficiency Analysis

We now provide a detailed latency comparison at 1024 × 1024 resolution on A6000 GPUs, including Qwen Image, T2I Copilot (Qwen-Agents), and SIDiffAgent under matched conditions. While SIDiffAgent introduces additional inference time per prompt due to structured refinement and editing, we clarify that this cost is offset in practice by reducing the need for repeated user-prompted regenerations and manual corrections. Combined with the strong gains in semantic alignment and visual fidelity, this analysis presents a more transparent quality-compute tradeoff.

### Added Multi-Dimensional Evaluation

In addition to the benchmarks already reported, we now include human preference studies, as well as GenEval and DPG Bench results. These demonstrate consistent improvements across multiple dimensions such as counting, color consistency, spatial alignment, and attribute reasoning.

---

> ### Author Response · Authors · 2025-12-01
> **Final Remarks by Authors 2**
>
> We sincerely thank all reviewers for their thoughtful comments and constructive feedback, which have significantly improved our work. In the revised manuscript, we have comprehensively addressed each concern, incorporating additional experiments and detailed discussions. The changes have been highlighted in blue for better readability.

---

### Meta-Review · Area_Chair_qjcQ · 2026-01-07

**Summary:**

This paper presents an agentic framework for text-to-image diffusion models that enables self-improvements.
The submission received negative reviews from the reviewers.
The reviewers mainly recognize the simplicity of the approach and good empirical performance.
The main concerns from the reviewers were the computation efficiency (all), limited novelty (eqU4, ZRAH), method being Qwen-specific (Bqk4, ZQuM), and limited evaluation (all).
After reading the paper, the reviewers' comments and the authors' rebuttal, the AC believes the authors' responses would have partially addressed the reviewers' concerns, especially on the experiments, but there would still be critical outstanding concerns regarding limited novelty, and lack of theoretical depth. The high computation overhead also remains an issue. The AC believes the remaining weaknesses would still outweigh the merits and does not recommend acceptance.

**Reviewer Concerns:**

Reviewers' concerns mostly addressed:
- Generalization to other models (Bqk4, ZQuM)

Partially addressed concerns:
- Missing comparisons and evaluation (eqU4, ZRAH, Bqk4, ZQuM)
- Failure case analysis (ZRAH)

Outstanding concerns:
- Limited novelty (eqU4, Bqk4)
- Lack of theoretical depth (Bqk4)
- Computation overhead (eqU4, ZRAH, Bqk4, ZQuM)

**Reviewer Scores:**

I think Bqk4's score would be raised to 6, while other reviewers would keep their original ratings.

---

### Decision · Program_Chairs · 2026-01-26

Reject